# 🐕 Both Ears Wide Open: TOWARDS LANGUAGE-DRIVEN SPATIAL AUDIO GENERATION

**Peiwen Sun**[*♠♡]**, Sitong Cheng**[*♡]**, Xiangtai Li**[*◇]**, Zhen Ye**[♡]**, Huadai Liu**[♣]**, Honggang Zhang**[♠]**, Wei Xue**[✉♡]**, Yike Guo**[✉♡]

♡ Hong Kong University of Science and Technology,
♠ Beijing University of Posts and Telecommunications,
◇ Nanyang Technological University, ♣ Zhejiang University
🌐 Website: https://peiwensun2000.github.io/bewo/

## ABSTRACT

Recently, diffusion models have achieved great success in mono-channel audio generation. However, when it comes to stereo audio generation, the soundscapes often have a complex scene of multiple objects and directions. Controlling stereo audio with spatial contexts remains challenging due to high data costs and unstable generative models. To the best of our knowledge, this work represents the **first attempt** to address these issues. We first construct a large-scale, simulation-based, and GPT-assisted dataset, **BEWO-1M**, with abundant soundscapes and descriptions even including moving and multiple sources. Beyond text modality, we have also acquired a set of images and rationally paired stereo audios through retrieval to advance multimodal generation. Existing audio generation models tend to generate rather random and indistinct spatial audio. To provide accurate guidance for Latent Diffusion Models, we introduce the **SpatialSonic** model utilizing spatial-aware encoders and azimuth state matrices to reveal reasonable spatial guidance. By leveraging spatial guidance, our model not only achieves the objective of generating immersive and controllable spatial audio from text but also extends to other modalities as the pioneer attempt. Finally, under fair settings, we conduct subjective and objective evaluations on simulated and real-world data to compare our approach with prevailing methods. The results demonstrate the effectiveness of our method, highlighting its capability to generate spatial audio that adheres to physical rules.

## 1 INTRODUCTION

The binaural hearing ability naturally enhances our perception of the world through the acoustic field, which became widely recognized in the 1980s with the rise of PCM (Lipshitz & Vanderkooy, 2004) and MP4 (Sikora, 1997) formats. In the current era of audio generation, creating immersive experiences requires the production of stereo audio that adheres to specific location properties, which can be effectively achieved through end-to-end generative models. This generation task boosts applications in immersive VR/AR (Fitria, 2023; Burdea & Coiffet, 2024) and embodied AI (Liu et al., 2024d). Therefore, generating stereo audio that incorporates spatial multimodal context represents a valuable task within the community.

Significant progress has been made in a monaural audio generation, with models such as AudioLDM 2 (Liu et al., 2024a), Make-an-Audio 2 (Huang et al., 2023a) and Tango 2 (Majumder et al., 2024). These models leverage the diffusion architecture to efficiently generate audio from textual prompts with a T5 model. For example, AudioLDM 2 uses a latent diffusion model to generate a latent representation of mel-spectrograms, and Make-an-Audio 2 and Tango 2 further explore the presence of events and their temporal ordering.

---

*: Equal contribution
✉: Corresponding authors

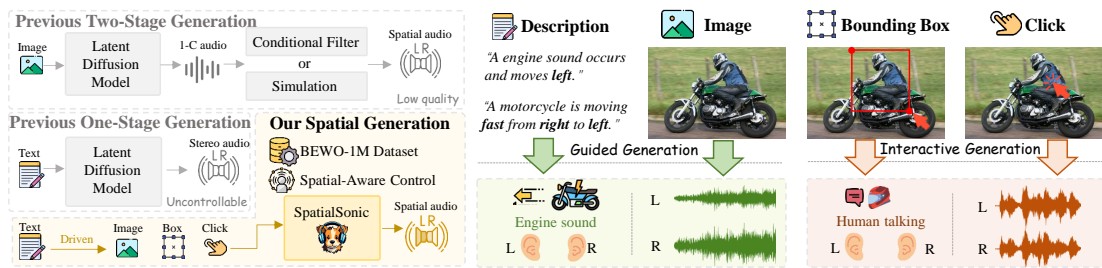

(a) Comparison to popular generation model    (b) The objective of controllable spatial audio generation

Figure 1: Our SpatialSonic, as a one-stage model, alleviates the problem of error accumulation in a two-stage model and facilitates control with end-to-end finetuning in a one-stage model. Moreover, our spatially enhanced system supports spatial audio generation from text and image, as well as interactive actions.

Compared to monaural audio generation, current research on spatial audio[1] generation is still limited. The pioneering work (Dagli et al., 2024) cascades the generation model with a simulator to generate stereo audio from the image. Another direct thought is converting generated mono audio to stereo by integrating interaural time difference (ITD) and interaural level difference (ILD) (Desai & Mehendale, 2022), which are crucial for spatial perception in human auditory processing. Typically, to convert mono audio to stereo, Zhou et al. (2020); Parida et al. (2022) use visual and positional cues as conditions to produce spatial audio through a U-net style filter. Although it seems feasible to generate spatial audio by cascading effective audio generation models with simulations or filters, the two-stage approach in Fig. 1(a) incurs high computational costs and potential error accumulation.

We attribute the challenges of spatial audio generation to 3 aspects: 1) large data scale; 2) precise guidance construction; 3) proper evaluation metrics. This work presents the **first exploration** addressing these issues.

Although stereo audio is common in real life, captions of such audio with spatial descriptions still require massive human resources. For example, generating spatial audio that matches the textual description "*a motorcycle engine sound moving gradually from front to left*" requires extensive paired data of motorcycles in various directions of movement. Due to such high labeling costs, the lack of sufficient high-quality data becomes a barrier to spatial audio generation models, compared to Mei et al. (2024); Wu et al. (2023). To better facilitate the advancement of multimodal guided spatial audio generation models, we have developed a dual-channel audio dataset named Both Ears Wide Open 1M (BEWO-1M). It contains up to **1 million audio samples** through rigorous simulations and GPT-assisted caption transformation. BEWO-1M contains an abundant soundscape, including moving-source, multi-source, and interleave-source scenarios with the spatial description or rational image. To ensure perceptual consistency with the real world, test sets from BEWO-1M are checked by humans, and a real-world recorded subset is manually constructed and annotated.

Previous one-stage stereo audio generation model (Evans et al., 2024a;b) training on real-world data is able to generate stereo audio based on the caption but fails to generate rational spatial audio, as shown in Fig. 1(a). After fine-tuning existing models with BEWO-1M, a basic capability can be obtained to understand the positional description. Since the knowledge from the text fine-tuned model is trained on our enormous data, it is important for the image to utilize this knowledge through language-driven behavior. In our diffusion-based **SpatialSonic** model, we first explore the multimodal spatial-aware guidance to encode images with regional perception way in Sec. 4.2. Then, we identify that due to the lack of explicit spatial guidance, simply fine-tuning the existing model with BEWO-1M still fails in precise T2A and I2A tasks. Therefore, we introduce azimuth guidance inducted by LLM through a specific scheme to clarify and integrate complex textual and visual contexts. Finally, along with proper classifier-free guidance training on the diffusion model, we obtain a controllable generation model. During inference, a straightforward method can be easily applied to achieve user interaction. We propose using ITD-based objective metrics and opinion-based subjective **evaluations**

---

[1]For clarity, we define "spatial audio" as the "stereo audio" adheres to spatial context.

to assess generated audio systematically. Our results show that SpatialSonic effectively generates realistic spatial audio, achieving a 70% reduction in ITD error and higher opinion scores than popular models with minor adaption, including AudioLDM2. Overall, our contributions are:

- Developing a semi-automated pipeline to create an open-source, large-scale, stereo audio dataset with spatial captions, **BEWO-1M** and supporting both large-scale training and precise evaluation.

- Introducing a one-stage, controllable, spatial audio generation framework, **SpatialSonic**, which is designed to generate dual-channel audio precisely adhering to multimodal spatial context.

- Proposing a series of **subjective and objective metrics** based on ITD and opinion score to evaluate spatial audio generation models. Under fair experimental conditions, our framework produces audio with enhanced spatial information, obtaining more authentic soundscapes.

## 2 RELATED WORKS

**Spatial Audio Understanding.** Researchers including May et al. (2010) have been exploring the field of stereo audio through learnable methods since 2010. In recent years, the development of deep learning has led to more extensive exploration of stereo audio. The first area explored is binaural audio localization, where researchers are able to localize the direction of sound sources by learning ITD and ILD in the single-source scenarios (Krause et al., 2023; Yang & Zheng, 2022; Cao et al., 2021; Shimada et al., 2022; Yasuda et al., 2020; García-Barrios et al., 2022) and the multi-source scenarios (Nguyen et al., 2020). With the advancement of multimodal research, mono-to-stereo audio generation methods conditioned on visual (Garg et al., 2021; Xu et al., 2021; Liu et al., 2024c; Li et al., 2024; Zhou et al., 2020), depth (Parida et al., 2022) and location (Leng et al., 2022), have also been gradually developed. The input takes a mono audio and an image, and then spatial audio can be obtained through supervised learning. However, since a mono signal is still required, this task is not a generation task from the current view. Additionally, researchers including Gebru et al. (2021); Phokhinanan et al. (2024); Ben-Hur et al. (2021); Geronazzo et al. (2020) hope to implicitly establish head-related transfer function (HRTF) through deep learning to form a reasonable signal mapping.

**Text-to-audio (T2A) Generation.** Audio generation from text is typically categorized into text-to-speech, text-to-music, and text-to-audio. In this paper, we focus on the latter category. From a methodological perspective, text-to-audio generation approaches can be broadly classified into autoregressive models and diffusion models. The pioneering works from Kreuk et al. (2022); Yang et al. (2023); Lu et al. (2024); Liu et al. (2024b) on autoregressive-based audio generation use a general audio tokenizer to convert waveforms into a sequence of tokens. Then, they apply an autoregressive network like GPT-2 (Radford et al., 2019) to predict the next token in the sequence. However, autoregressive-based models often require substantial data and computational resources (Kreuk et al., 2022). This has led to increased exploration of diffusion-based models for audio generation, with outstanding works including AudioLDM (Liu et al., 2023; 2024a), Audiobox (Vyas et al., 2023) and Stable Audio (Evans et al., 2024a;b).

While the controllable problem in image generation (Cao et al., 2024) persists in audio, the specific challenges may differ. General controlling attempts are made by enhancing the text prompt (Copet et al., 2024; Huang et al., 2023b). To improve the precision of time and frequency control, Huang et al. (2023a); Xie et al. (2024b) explore the temporal encoding along with the fusion mechanics. As for stereo audio, MusicGen (Copet et al., 2024) and Stable Audio (Evans et al., 2024a;b) can generate stereo audio but without spatial control. In this work, we reveal and deal with this spatial-controlling problem for the first time.

**Other-to-audio Generation.** With ImageHear (Sheffer & Adi, 2023) pioneering the use of images as guidance, CLIP has been extensively employed for I2A generation tasks. Subsequently, Dong et al. (2023) and Wang et al. (2024) have also leveraged CLIP to generate realistic spectra through diffusion guidance directly. However, the CLIP primarily focuses on aligning the global abstract semantics rather than the positional context. Therefore, spatial audio generation, as a position-aware task, requires further exploration of regional understanding of images.

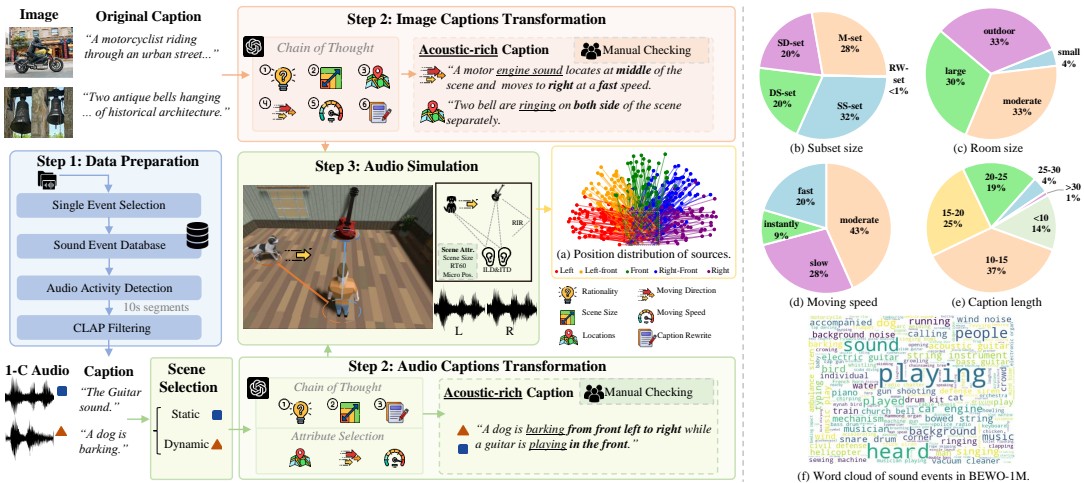

Figure 2: The pipeline of BEWO-1M data collection. The data machine is driven by LLM induction and rigorous simulation. In particular, the data for testing are built with human checking. The diagram in Step 3 represents one of the simulation scenarios. (a) illustrates the diversity of source and microphone positions. (b-f) show the abundant soundscapes in BEWO-1M.

## 3 BEWO-1M DATASET

Due to high labeling costs, Tab. 1 shows that stereo audio datasets usually have a short duration. The limited spatial audio datasets and the spatial constraints in existing models (Evans et al., 2024b) necessitate the creation of a dataset with explicit spatial context. We propose BEWO-1M, a large-scale stereo audio dataset with spatial captions, as the first to the best of our knowledge. BEWO-1M consists of audio-caption pairs and audio-image pairs. The construction pipeline in Fig. 2 follows the step-by-step procedure of preparing, transforming, and simulating to obtain a balanced dataset.

**Data Preparation**. Raw data collected online is noisy and requires pre-selection before simulation. Initially, we select samples with captions describing single sound sources to construct a single-source database. This makes each audio clip contain only a single sound event, thereby ensuring realism and quality in simulation. We then apply sound activity detection on each audio and randomly crop to $10s$ segments. Further, we remove segments with low CLAP (Wu et al., 2023) similarity with their caption. See Appendix B.1 for more details.

Table 1: Comparison of existing audio-caption datasets.

| Task | Dataset | Duration (hours) | Num. of Audios | Paired Type |
|---|---|---|---|---|
| Event | LAION-Audio (Wu et al., 2023) | 4.3k | 633k | Text |
| | WavCaps (Mei et al., 2024) | 7.5k | 403k | Text |
| | AudioCaps (Kim et al., 2019) | 110 | 46k | Text |
| | SoundDescs (Koepke et al., 2022) | 1.1k | 33k | Text |
| | Clotho (Drossos et al., 2020) | 23 | 25k | Text |
| | Audio Caption (Wu et al., 2019) | 10.3 | 3.7k | Text |
| | VGG-Sound Chen et al. (2020) | 550 | 200k | Video |
| | AVE Tian et al. (2018) | 11.5 | 4k | Video |
| Temporal | PicoAudio (Xie et al., 2024b) | 15.6 | 5.6k | Text |
| | AudioTime (Xie et al., 2024a) | 15.3 | 5.5k | Text |
| | CompA-order (Ghosh et al., 2024) | 1.5 | 851 | Text |
| Spatial | SimBinaural (Garg et al., 2023) | 116 | 22k | Video |
| | FAIR-Play (Gao & Grauman, 2019) | 5.2 | 1.9k | Video |
| | YT-ALL (Morgado et al., 2018) | 113.1 | 1.1k | Video |
| | MUSIC (Zhao et al., 2018) | 23 | 0.7k | Video |
| | BEWO-1M (Ours) | 2.8k | 1,016k | Text |
| | BEWO-1M (Ours) | 54 | 2.3k | Image |

**GPT-based Attributes Induction and Caption Transformation.** With input as a caption or image-caption pair, we use GPT-4 and GPT-4o to induct sounding objects and their attributes for simulation, transforming raw captions into acoustic-rich captions with spatial descriptions. The audio simulator normally requires certain essential attributes to simulate realistic audio, including scene size, sound source location, moving direction, and speed. We create an audio-object pool from sound objects inducted from audio-caption pairs. The sound objects are inducted from images, and then the corresponding audio is retrieved from this pool. To obtain reasonable attributes and captions, we apply the Chain of Thought Prompting (CoT) (Wei et al., 2023) to enhance the induction

ability of GPT-4 and GPT-4o. We predefined several patterns to describe each attribute, with each being an attribute element, such as "far" or "near" for the distance attribute. Specifically, we require GPT to select attribute labels that match the input context and transform raw captions into acoustic-rich captions with positional and movement phrases. Fig. 2(e) presents the statistics of caption length in BEWO-1M, and the transformed captions still remain concise with additional spatial and movement descriptions. See Appendix B.2 for more details.

**Audio Simulation.** We utilize the obtained attributes to simulate realistic and reasonable stereo audio. Following prior researches (Salvati et al., 2021; Chen et al., 2022b; Dagli et al., 2024), we use Pyrooma-coustics (Scheibler et al., 2018) and gpuRIR (Diaz-Guerra et al., 2021) for simulation. To enhance diversity and reflect real-world distribution, we introduce a certain level of randomness into the inferred and selected attributes. To ensure scenery diversity, we also randomly set additional scene attributes like the microphone position and the room reverberation indicator, RT60. The simulation assumes a common ear distance of 16∼18 cm. To make the dataset general, we do not consider the shadow effect of the head and leave the head adaptation achieved by future fine-tuning. The simulator then uses these attributes to generate the audio. Fig. 2(a) shows the diversity of source positions in simulation. In the indoor scene, the simulator also generates audio with a static simulated room impulse response (RIR). For the moving source scenarios, we build a trajectory to detail its positions. Our pipeline proficiently simulates audio across various environments, achieving both diversity and authenticity while meeting the criteria for ITD and ILD. See Appendix B.4 for details.

**Post-Processing.** To ensure data quality, we perform manual checking for part of the training set and entire test set. Tab. C14 shows that our automated pipeline can generate decent captions.

In summary, we constructed 2.8k hours of training audio with more than 1M audio-text pairs and approximately 17 hours of validation data with 6.2k pairs. As shown in Tab. 1, our dataset, as the first audio-caption dataset with stereo audio and spatial descriptions, is comparable to other monaural audio-caption datasets. In addition to its large scale, the BEWO-1M dataset is also notable for its high quality. As shown in Tab. 4, the simulated audio receives high subjective ratings from human annotators. Fig. 2(c, d) presents the diversity of attributes in BEWO-1M, and Fig. 2(f) shows the sound events diversity. The details and

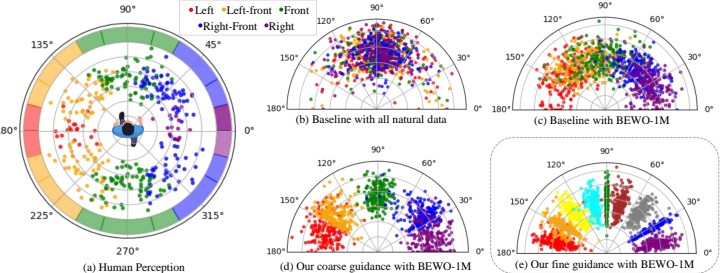

Figure 3: In polar coordinates, the radius represents normalized audio energy, the angle denotes the perception angle (0°for right, 180°for left), and five colors in the above legend signify the use of common directional terms to describe sound events. (a) is the human perception based on the questionnaire of volunteers. In (b), the baseline fails to generate the controllable audio. Obviously, (c) highlights the valuable knowledge from BEWO-1M. (d,e) highlights the superiority of our data and methods in controlling the generation of 5 common directions and uniform fine-controlling matrices.

statistics of subsets are provided in Appendix C. Finally, we fine-tune the baseline (Evans et al., 2024b) with BEWO-1M, and Fig. 3(b,c) demonstrates our dataset significantly improves the spatial discrimination.

## 4 METHOD

**Overview.** Our objective is to extract precise guidance from multimodal input and create stereo audio adhering to spatial context. The overall pipeline of our SpatialSonic network consists of multimodal extraction, azimuth guidance, and latent code generation, as illustrated in Fig. 4. On top of Evans et al. (2024b), we introduce a multimodal encoder for the image's spatial perception in Sec. 4.2, to adapt the image to T2A

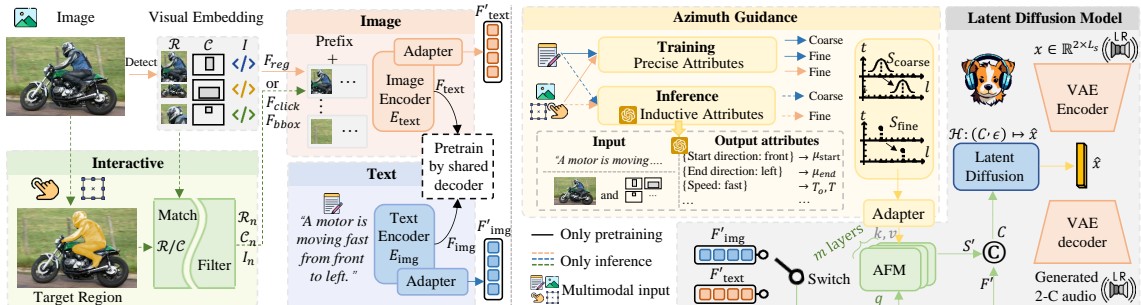

Figure 4: The overall pipeline of SpatialSonic. It is a one-stage controllable model that processes multimodal inputs to generate spatial audio, where GPT is used to inject the specific azimuth state into the guidance.

model. Then, the azimuth fusion module in Sec. 4.3 provides extra clear conditions with the help of LLM and the azimuth scheme. Finally, the training and inference methodologies are presented in Sec. 4.4.

## 4.1 PRELIMINARIES

**Task Objective:** Let $x \in \mathbb{R}^{2 \times L_s}$ represent a dual-channel audio signal, where $L_s$ depends on the length of audio. As the autoencoder compresses the $x$ into $\hat{x}$, the audio generation process can be denoted as $\mathcal{H}: (C, \epsilon) \mapsto \hat{x} \mapsto x$, where $C$ is the multimodal condition, $\epsilon$ is the Gaussian noise and $\mathcal{H}$ is the conditional generation process.

**Text Embedding:** The pre-trained language model T5 encoder (Raffel et al., 2023) is used as the text encoder ($E_{\text{text}}$). It captures spatial context as text embedding $F_{\text{text}} \in \mathbb{R}^{L \times d_{\text{text}}}$, where $L$ is a variable number depends on the text input. The ablation study in Tab. 7 reveals that incorporating CLAP speeds up convergence compared to T5; however, the spatial performance is inferior to T5.

**Image Embedding:** Previous image encoder $E_{\text{img}}$ used in I2A usually obtains $F_{\text{img}} \in \mathbb{R}^{d_{\text{img}}}$ by CLIP. Considering the regional perception in Sec. 4.2, our image embedding follows $F_{\text{img}} \in \mathbb{R}^{L \times d_{\text{img}}}$.

**Azimuth Information:** The attributes including number of sources $K$, start position $\mu_{\text{start}}$, end position $\mu_{\text{end}}$, moving start time $T_0$ and moving interval $T$ is precisely known during simulation and can be used during training. During inference, inspired by Xie et al. (2024b); Qu et al. (2023), $K, \mu_{\text{begin}}, \mu_{\text{end}}, T$ can be inducted by GPT. Details are provided in Appendix H.

## 4.2 IMAGE EMBEDDING WITH REGIONAL PERCEPTION

Popular I2A model (Sheffer & Adi, 2023) using CLIP (Ramesh et al., 2022) focuses on aligning the global abstract semantics rather than its positional and relational context. Therefore, the objective of this module is to gain regional image perception, while re-utilizing language-driven knowledge from massive text pairs.

To inject the spatial-aware semantics into the image encoder, we follow Cho et al. (2021) to carry out a detection model as a regional perception to provide the rich positional context; this pre-training network is an encoder-decoder structure, firstly trained as a vision-language task. The network generates the acoustic description of the image, supervised by image-caption pairs in Sec. 3. The pre-training follows 3 steps below. 1) Use the detection model to obtain the $N$ regions' embedding $\mathcal{R} \in \mathbb{R}^{N \times L_{\text{clip}}}$, corresponding coordinates $\mathcal{C} \in \mathbb{R}^{N \times 4}$, and detected object class embedding $\mathcal{I} \in \mathbb{R}^{N \times L_{\text{cls}}}$, where $0 \leq N \leq N_{\text{max}}$. Obtain a visual embedding $F_{\text{reg}} \in \mathbb{R}^{N \times L_{\text{reg}}}$ by linear projecting and element-wise adding $\mathcal{R}, \mathcal{C}, \mathcal{I}$ separately. 2) Concatenate the visual embedding to specified text prefix embedding (i.e. "*image acoustic captioning*") as encoder input. 3) Update the multimodal encoder while keeping the weight of the decoder frozen. Overall, since the shared decoder is utilized during the pre-training of text and visual encoders, the latent features obtained from the

visual encoder $E_{\text{img}}$ and text encoder $E_{\text{text}}$ can be regarded in the same aligned space. The visual embedding output by $E_{\text{img}}$ can be denoted as $F_{\text{img}} \in \mathbb{R}^{L \times d_{\text{img}}}$.

## 4.3 Controlling Azimuth from Coarse to Fine

When text and image embedding are used directly as conditions, there is still a large dispersity in Fig. 3(c). Therefore, it is crucial to design a model that effectively supports precise generation using both text and images. On the one hand, the language phrase to describe source direction is limited and rather subjective. By learning the underlying distribution of human perception in Fig. 3(a), the generated audio becomes more realistic in assessments. On the other hand, when generating stereo audio using continuous spatial information such as images, it is necessary to form a more precise guidance. Inspired by layout-controllable image generation (Inoue et al., 2023; Qu et al., 2023), we introduce simple and clear guidance, azimuth state matrix for $K$ sources $S \in \mathbb{R}^{K \times L_{\text{azi}} \times d_{\text{time}}}$, which encodes azimuth at different times slots.

To fit the coarse guidance into a distribution of human perception, we developed a Gaussian-based coarse-grained guidance for controllable generation. Given a time and duration $\{t, T\} \in d_{\text{time}}$ based on specified speeds, the center position $\mu(t)$ at given moment $t$ adheres to the following physical principles:

$$\mu(t) = \mu_{\text{start}} + \frac{t}{T}(\mu_{\text{end}} - \mu_{\text{start}}). \tag{1}$$

where $\mu_{\text{start}}, \mu_{\text{end}} \in L_{\text{azi}}$. Here, for conciseness, we omit the expression before moving ($t < T_0$) and after moving ($T_0 + T < t$), which can be simply derived by $\mu_{\text{end}} = \mu_{\text{start}}$. Further, the azimuth of $k$-th ($0 \le k < K$) object under the normal distributions $\mathcal{N}(\cdot)$ at different moments are illustrated by,

$$S_{l,t}^{\text{coarse}}[k] = \mathcal{N}(l \mid \mu(t), \sigma^2) = \frac{1}{\sqrt{2\pi\sigma^2}} \exp\left(-\frac{(l - \mu(t))^2}{2\sigma^2}\right), \tag{2}$$

where the variance $\sigma$ is obtained from the statistics of real data. Each azimuth $l$ can be modeled as $l = 1$ for right and $l = d_{\text{time}}$ for left. Then $S_{l,t}^{\text{coarse}}[k]$ is azimuth-wise normalized before the next module.

For fine-grained purposes, the precise location is easily accessible to the nature of the simulation. Thus, we design the discrete state matrix of $k$-th object to represent the precise azimuth across time as

$$S_{l,t}^{\text{fine}}[k] = \begin{cases} 1 & \text{if } l = \lfloor \mu(t) \rfloor \\ 0 & \text{otherwise} \end{cases}. \tag{3}$$

This fine matrix can be regarded as the extreme situation of equation 2 without the uncertainty.

Then, we enhance the condition by fusing azimuth state and text embedding by azimuth fusion module (AFM). This AFM composes of the multi-layer cross-attention as $\text{CA}(q, k, v)$ introduced as

$$\begin{aligned} S' &= \text{CA}(S, F'_m, F'_m) + S \\ &= \text{softmax}(S \cdot F'_m)F'_m + S, \end{aligned} \quad \text{for } F'_m = \begin{cases} F'_{\text{text}} & \text{if T2A} \\ F'_{\text{image}} & \text{if I2A} \end{cases}, \tag{4}$$

where $F'_{text}, F'_{image}$ are the embedding after transforming $F_{text}, F_{image}$ by adapter and $S'$ is the attended state matrix. To enhance conciseness, the equation 4 for attention have excluded projection and scaling factors. This state matrix and the fusion module allow us to encode complex behaviors such as the speed and direction of sound source movement, and even support future custom matrices for any speed and azimuth.

## 4.4 Training and Inference of Diffusion Model

We utilize a diffusion model as $\mathcal{H}$ to model $\hat{x}$ based on the azimuth state matrix $S$ and modality embedding $F_{\text{text}}$ or $F_{\text{image}}$. The forward process of the cosine form (Esser et al., 2024) is used to obtain the noised representation $\mathcal{P}_\tau$ of each time step $\tau$ by noise $\epsilon$ injection by

$$\mathcal{P}_\tau = \alpha\hat{x} + \beta\epsilon, \tag{5}$$

where $\epsilon$ follows a isotropic Gaussian distribution, $\alpha = \cos\left(\frac{\pi}{2}\tau\right)$ and $\beta = \sin\left(\frac{\pi}{2}\tau\right)$. Then, the v-prediction parameterization (Kingma & Gao, 2024) is implemented for training for better sampling stability with a few of the inference steps, so the overall training conditional objective is

$$\mathcal{L} = \mathbb{E}_{t\sim[0,1],\sigma_\tau,\mathbf{x}_{\sigma_\tau}} w_t \left\| f_\theta\left([\mathcal{P}_\tau, cat(S', F'_m)], \sigma_\tau\right) - v_{\sigma_\tau} \right\|_2^2, \tag{6}$$

$$v_{\sigma_\tau} = \frac{\partial \mathcal{P}_\tau}{\sigma_\tau} = \alpha\epsilon - \beta\mathcal{P}_0, \tag{7}$$

where the velocity $v$ is calculated from noise schedule $\sigma_\tau \in [0, 1]$, and $w_\tau$ is the weight obtained from the signal-to-noise ratio of $\mathcal{P}_\tau$. $cat(\cdot, \cdot)$ means concatenation. $f_\theta$ is the estimation network to reconstruct and denoise from $\mathcal{P}_\tau$, to learn the conditional denoising as

$$\hat{x} = \frac{1}{\alpha}[f_\theta\left(CA\left(\mathcal{P}_\tau, cat(S', F'_m)\right)\right) - \beta\epsilon]. \tag{8}$$

Initially, the T2A model is trained using the BEWO-1M dataset. On top of this T2A model, it is fine-tuned using the spatial-aware image encoder to develop the I2A model. Drawing inspiration from current SAM-based interaction models (Ma et al., 2024; Kirillov et al., 2023), we utilize clicks and bounding boxes (BBox) to select Region of Interest (RoI) from regions $\mathcal{R}_n \in \mathcal{R}$ in Sec. 4.2. After generating the high-quality mask, it is then matched with region coordinates $\mathcal{C}$ and regional feature $\mathcal{R}$. We enable the selection of multiple RoIs as input of $E_{\text{image}}$. The regions $\mathcal{R}_n$, coordinates $\mathcal{C}_n$ and IDs $\mathcal{I}_n$ are selected, where $n \in N$. By using the selected $\mathcal{R}_n$, $\mathcal{C}_n$ and $\mathcal{I}_n$ as input, the image encoder $E_{\text{img}}$ takes interactive embedding $F_{\text{click}}$ or $F_{\text{bbox}}$ and finally generate the spatial audio of the target object.

## 5 EXPERIMENT

### 5.1 TRAINING AND EVALUATION

**Dataset** Our dataset is built on a diverse combination of datasets detailed in Appendix B.1. We convert the sampling rate of audios to 16kHz and pad short clips to 10 seconds long after the data construction in Sec. 3. As for images, we select the scenery with audible subjects from COCO2017 (Lin et al., 2014).

**Model Configurations** We fine-tune the continuous autoencoder pre-trained by Stability AI[2] to compress the perceptual space with downsampling to the latent representation. For our main experiments, we train a text-conditional Diffusion-Transformer (DiT) (Levy et al., 2023; Peebles & Xie, 2023), which is optimized using 8 NVIDIA RTX 4090 GPUs for 500K steps. The base learning rate is set to $2e$-5 with a batch size of 128 and audio length of $10s$. Hyper-parameters are detailed in the Appendix E.

**Evaluation Metrics:** 1) 1-C metrics: We adopt metrics from Huang et al. (2023b) and Liu et al. (2023) to calculate Fréchet Distance (FD), Inception Score (IS), Kullback-Leibler divergence (KL), Fréchet Audio Distance (FAD), CLAP score (CLAP), overall impression (OVL) and audio-text relation (REL) for T2A evaluation. Additionally, CLIP Score (Wu et al., 2022; Sheffer & Adi, 2023) evaluates the relevance for I2A model. 2) 2-C objective metric: To better examine the quality of the generated spatial audio, we propose novel evaluation methods based on TDOA[3]. Utilizing the non-silent segments with a threshold of -16 dBFS in the audio, we compute TDOA distributions in intervals of 0.1 seconds using both the traditional Generalized Cross-Correlation with Phase Transform (GCC-PHAT) (Knapp & Carter, 1976) and the deep learning network StereoCRW (Chen et al., 2022b). Finally, the mean absolute error is computed based on the TDOA of the ground truth and generated audio as **GCC MSE** and **CRW MSE**. A lower error indicates better alignment with ground truth, but simple MSE fails in scenarios with multiple or moving sources. Expanding on FAD (Kilgour et al., 2018), a novel evaluation metric, Fréchet Stereo

---

[2]https://github.com/Stability-AI/stable-audio-tools

[3]This quantity is known as both difference of arrival (TDOA) and the interaural time difference (ITD).

Audio Distance (**FSAD**), is introduced. FSAD builds on FAD by leveraging the StereoCRW network instead of VGGish for FD computation. More methodologies and parameters are detailed in Appendix F.2. 3) 2-C subjective metric: To assess the quality of the generated audio, we employ the Mean Opinion Score (MOS) through human evaluation to evaluate the quality of source direction (MOS-Direction) and event (MOS-Event) separately. We invite 15 experts to evaluate the quality on a scale ranging from 1 to 5, with 5 for the best quality. For further information on our MOS, please refer to Appendix F.4.

## 5.2 RESULTS

**Comparison on 1-C T2A Baseline:** After averaging the generated stereo audio across channels, we compute metrics for mono-channel (1-C) audio. The results are presented in Tab. 2 and highlight our model's strong performance across all metrics, particularly excelling in IS and CLAP, often matching or even surpassing best benchmarks.

Table 2: 1-C audio quality comparison of popular methods in T2A on Audiocaps test set.

| Model | Objective | | | | | Subjective | |
|---|---|---|---|---|---|---|---|
| | FD↓ | IS↑ | KL↓ | FAD↓ | CLAP↑ | OVL↑ | REL↑ |
| AudioGen-L (Kreuk et al., 2022) | - | - | 1.69 | 1.82 | - | / | / |
| Uniaudio (Yang et al., 2023) | - | - | 2.60 | 3.12 | - | 3.05 | 3.19 |
| Make-An-Audio (Huang et al., 2023b) | 18.32 | 7.29 | 1.61 | 2.66 | 0.539 | 3.53 | 3.59 |
| Make-An-Audio 2 (Huang et al., 2023a) | **11.75** | 11.16 | 1.32 | **1.80** | 0.645 | 3.72 | 3.57 |
| AudioLDM (Liu et al., 2023) | 23.31 | 8.13 | 1.57 | 1.96 | 0.621 | 3.70 | 3.71 |
| AudioLDM 2 (Liu et al., 2024a) | 19.93 | 9.39 | 1.64 | 1.86 | 0.652 | **3.78** | **3.76** |
| Stable-audio-open (Evans et al., 2024b) | 21.21 | 10.50 | 1.86 | 2.37 | 0.594 | 3.64 | 3.60 |
| TANGO (Ghosal et al., 2023) | 26.13 | 8.23 | 1.37 | 1.83 | 0.650 | 3.65 | 3.66 |
| TANGO 2 (Majumder et al., 2024) | 18.85 | 10.09 | **1.12** | 1.90 | **0.675** | 3.73 | 3.69 |
| SpatialSonic (Ours) | 14.03 | **13.79** | 1.37 | 1.93 | 0.672 | 3.75 | 3.73 |

These results indicate that our model adeptly interprets textual cues to produce high-quality audio outputs.

Table 3: 2-C audio quality comparison of popular methods in T2A on BEWO-1M test set. "†" means a minor adjustment of the original structure with a conditional mono-to-stereo filter.

| Task | Method | Objective | | | Subjective | |
|---|---|---|---|---|---|---|
| | | GCC MSE ↓ | CRW MSE ↓ | FSAD ↓ | MOS-Events ↑ | MOS-Direction ↑ |
| Simulation | Simulation | - | - | - | 4.94 | 4.95 |
| T2A (SS-set) | AudioLDM2† | 46.59 | 50.17 | 1.61 | 3.57 | 3.53 |
| | Make-An-Audio2† | 38.83 | 43.12 | 0.97 | 3.58 | 3.59 |
| | Stable-audio-open | 38.73 | 34.36 | 0.63 | 3.73 | 3.76 |
| | SpatialSonic(Ours) | **27.20** | **15.86** | **0.17** | **3.78** | **3.84** |
| T2A (SD-set) | AudioLDM2† | 45.08 | 42.88 | 0.94 | 3.37 | 3.34 |
| | Make-An-Audio2† | 48.55 | 47.88 | 1.09 | 3.38 | 3.30 |
| | Stable-audio-open | 45.76 | 48.60 | 0.53 | 3.68 | 3.58 |
| | SpatialSonic(Ours) | **44.36** | **31.91** | **0.26** | **3.86** | **3.71** |
| T2A (DS-set) | AudioLDM2† | 38.96 | 50.96 | 2.48 | 3.29 | 2.97 |
| | Make-An-Audio2† | 35.37 | 48.54 | 2.11 | 3.24 | 3.31 |
| | Stable-audio-open | 32.63 | 36.30 | 0.87 | 3.60 | 3.61 |
| | SpatialSonic(Ours) | **22.51** | **13.75** | **0.31** | **3.80** | **3.83** |
| T2A (M-set) | AudioLDM2† | 36.43 | 49.87 | 1.22 | 3.38 | 3.34 |
| | Make-An-Audio2† | 36.01 | 47.31 | 1.32 | 3.29 | 3.32 |
| | Stable-audio-open | 34.20 | 48.06 | 0.53 | 3.54 | 3.58 |
| | SpatialSonic(Ours) | **33.32** | **43.24** | **0.16** | **3.75** | **3.73** |
| T2A (RW-set) | AudioLDM2† | 46.98 | 44.66 | 1.54 | 3.28 | 3.35 |
| | Make-An-Audio2† | 46.94 | 43.47 | 1.60 | 3.23 | 3.30 |
| | Stable-audio-open | 43.18 | 47.58 | 0.60 | 3.51 | 3.49 |
| | SpatialSonic(Ours) | **30.27** | **23.19** | **0.28** | **3.79** | **3.76** |

**Comparison on 2-C T2A Baseline:** Based on the proposed evaluation method, we train all the models listed on the BEWO-1M train set and conduct a series of tests on spatial audio in Tab. 3. 1) The authenticity of our ground truth simulation data is evaluated by humans, showing that our simulation received a high position and events score. 2) The evaluation metrics GCC MSE and CRW MSE, based on global averages, show limitations in representing complex subsets such as the SD-set and DS-set; thus, we primarily treat FSAD as our evaluation method. 3) Utilizing FSAD and MOS as the principal metric, our approach outperforms all baselines in objective performance and achieves higher recognition in subjective evaluations.

**Comparison on 2-C I2A Baseline:** As the objects in COCO2017 often have common occlusion, our method still demonstrated advantages in objective and subjective evaluation metrics, notably achieving a 1.56 performance improvement in FSAD. Additionally, we extend our tests to earlier I2A datasets, which also show the superiority of SpatialSonic in the Appendix G.3.

Table 4: 2-C audio quality comparison of popular methods in I2A generation. "S&H" means Seeing-and-Hearing.

| Task | Method | Objective | | | | Subjective | |
|---|---|---|---|---|---|---|---|
| | | CLIP Score ↑ | GCC MSE ↓ | CRW MSE ↓ | FSAD ↓ | MOS Events ↑ | MOS Direction ↑ |
| GT | Simulation | 6.241 | - | - | - | 4.61 | 4.68 |
| V2A | See2sound | 1.410 | 97.90 | 60.73 | 2.49 | 3.09 | 3.17 |
| | S&H | 4.737 | - | - | - | 3.53 | - |
| | S&H† | 4.591 | 96.11 | 62.55 | 2.08 | 3.39 | 3.47 |
| | SpatialSonic | **5.618** | **80.20** | **57.37** | **0.52** | **3.68** | **3.79** |

Table 5: 2-C audio quality comparison in an interactive I2A generation.

| Prompt | Method | Subjective | | Clarity | |
|---|---|---|---|---|---|
| | | MOS Events ↑ | MOS Direction ↑ | GCC MA↑ | CRW MA↑ |
| BBox | See2sound | 3.55 | 3.47 | 1.91 | 8.11 |
| | S&H† | 3.44 | 3.31 | 2.91 | 10.37 |
| | SpatialSonic | **3.68** | **3.64** | 14.60 | 17.32 |
| Point | See2sound | 3.47 | 3.50 | 1.02 | 7.11 |
| | S&H † | 3.26 | 3.43 | 2.77 | 8.91 |
| | SpatialSonic | **3.58** | **3.61** | 15.91 | 18.11 |

**Comparison on 2-C Interactive I2A Baseline:** Given the absence of a ground truth audio for our interactive objective, we construct a small comprising around 150 images, 300 bounding boxes, and 300 click points derived from authentic user interactions (accessible in the Appendix C.3). We then evaluate the generation quality across two dimensions using subjective metrics. Additionally, the clarity of direction is reflected by calculating the mean absolute ILD as GCC MA and CRW MA.

Table 6: Ablation study of the coarse and fine strategy on different subsets.

| Training | Inference | FSAD @M-set ↓ | FSAD @RW-set ↓ | FSAD @I2A ↓ |
|---|---|---|---|---|
| $w/o$. Coarse & Fine | $w/o$. Matrix $S$ | 0.53 | 0.60 | 1.41 |
| $w/$. Coarse & Fine | $w/o$. Matrix $S$ | 0.56 | 0.67 | 0.84 |
| | $w/$. Fine $S$ | 0.32 | 0.48 | **0.52** |
| | $w/$. Coarse $S$ | **0.16** | **0.28** | 0.96 |

Table 7: The ablation study on the textual guidance with 20% of training data BEWO-1M.

| Text Encoder | Converge Iter. | CLAP @Audiocaps ↓ | FSAD @M-set ↓ |
|---|---|---|---|
| $w/$. CLAP | 80k | 2.33 | 0.56 |
| $w/$. T5 | 135K | 2.35 | **0.24** |
| $w/$. T5+CLAP | 105K | **2.31** | 0.40 |

**Ablation on Each Component:** 1) Text encoder: Just as stated in Sec. 4.1, although T5 converges slowly, it has significant advantages in capturing spatial and temporal information, as shown in Tab. 7. 2) Coarse and fine azimuth matrix: The results in Tab. 6 indicate that coarse guidance is more suitable for the T2A task, while fine guidance is necessary for the T2A task. Quantitatively, using fine guidance in T2A causes too strict azimuth guidance, even for noise, whereas coarse guidance in I2A results in a less unstable generation for complex scenarios.

**More Experiments:** Notably, extensive statistics, detailed experiments, and comprehensive user studies are presented in the appendix. More insights in Appendix G are strongly recommended to readers for a thorough understanding of the challenges, including how azimuth guidance and caption length affect audio quality.

## 6 DISCUSSION AND CONCLUSION

**Discussion**: It is believed that BEWO-1M is able to facilitate widespread application in various areas such as 1) spatial cross-modal retrieval, 2) contrastive language-audio pre-training, 3) spatial audio captioning, 3) large-scale audio-language pertaining model. From a methodology perspective, our SpatialSonic model represents a pioneering effort to achieve controllable spatial audio generation. However, there is still potential for improvement. For example, the current image encoder's limited size restricts its ability to fully comprehend the dynamics and behaviors across datasets with more diverse classes or in open-world scenarios.

**Conclusion**: In this work, we introduce a novel task to generate stereo audio from spatial context, which requires the machine to understand multimodal information and generate rational stereo audio. To advance this field, We develop the first open-source, large-scale spatial audio dataset, BEWO-1M, for training and evaluation. Our proposed SpatialSonic model further establishes a robust baseline with enhanced spatial perception. During experiments, we compare our SpatialSonic with several existing models training on BEWO-1M and demonstrate SpatialSonic's promising performance in generating high-quality stereo audio adhering to spatial locations. Our task and dataset have great potential in applications such as AR/VR and embodied AI to create immersive experiences. In the future, we plan to further expand the scale of the current dataset with 5.1-channel audios, a higher sampling rate, and more visual data, including images and videos, to meet the growing data demands in the era of generation.

# 7 ACKNOWLEDGEMENTS

The research was supported by NSFC (No. 62206234), Early Career Scheme (ECS-HKUST22201322), Theme-based Research Scheme (T45-205/21-N) from Hong Kong RGC, and Generative AI Research and Development Centre from InnoHK

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

# A  COMPARISON WITH TO PREVIOUS WORK

Compared to the generation tasks supported by other models, our model not only supports T2A and I2A as shown in Tab. A8 but also enables controllable spatial audio generation. Overall, our model mainly supports various related generation tasks based on spatial audio.

Table A8: Comparison of audio generation methods. "T2A" means text-to-audio. "I2A" means image-to-audio. "V2A" means video-to-audio. "A2A" means audio-to-audio (audio impainting, style transfer...). ✓ means "support". ✗ means "not supported". ◐ means "partial support"; although the networks were not specifically designed for this purpose, their model demonstrates some capability to meet the demand.

| | Supported Modality | | | | | Controllable Features | | |
|---|---|---|---|---|---|---|---|---|
| | T2A | I2A | V2A | A2A | interactive2A | temporal | stereo | spatial |
| audiogen (Kreuk et al., 2022) | ✓ | ✗ | ✗ | ✗ | ✗ | ✗ | ✗ | ✗ |
| uniaudio (Yang et al., 2023) | ✓ | ✗ | ✗ | ✓ | ✗ | ✗ | ✗ | ✗ |
| Unified-IO 2 (Lu et al., 2024) | ✓ | ✗ | ✗ | ✗ | ✗ | ✗ | ✗ | ✗ |
| Make-an-audio (Huang et al., 2023b) | ✓ | ◐ | ◐ | ✓ | ✗ | ◐ | ✗ | ✗ |
| Make-an-audio 2 (Huang et al., 2023a) | ✓ | ✗ | ◐ | ✗ | ✗ | ✓ | ✗ | ✗ |
| AudioLDM (Liu et al., 2023) | ✓ | ✗ | ◐ | ✓ | ✗ | ✗ | ✗ | ✗ |
| AudioLDM 2 (Liu et al., 2024a) | ✓ | ◐ | ◐ | ✓ | ✗ | ✗ | ✗ | ✗ |
| TANGO (Ghosal et al., 2023) | ✓ | ✗ | ✗ | ✗ | ✗ | ✗ | ✗ | ✗ |
| TANGO 2 (Majumder et al., 2024) | ✓ | ✗ | ✗ | ✗ | ✗ | ✓ | ✗ | ✗ |
| PicoAudio (Xie et al., 2024b) | ✓ | ✗ | ✗ | ✗ | ✗ | ✓ | ✗ | ✗ |
| Amphion (Zhang et al., 2023) | ✓ | ✗ | ✗ | ✗ | ✗ | ✓ | ✗ | ✗ |
| Seeing-and-hearing (Xing et al., 2024) | ✓ | ✓ | ✓ | ✗ | ✗ | ✗ | ✗ | ✗ |
| V2A-Mapper (Wang et al., 2024) | ✗ | ✓ | ✓ | ✗ | ✗ | ✗ | ✗ | ✗ |
| audiobox (Vyas et al., 2023) | ✓ | ✗ | ✗ | ✓ | ✗ | ✗ | ✗ | ✗ |
| stable-audio-open (Evans et al., 2024b) | ✓ | ✗ | ✗ | ✗ | ✗ | ◐ | ✓ | ◐ |
| SpatialSonic(Ours) | ✓ | ✓ | ✗ | ✗ | ✓ | ◐ | ✓ | ✓ |

# B  DATASET CONSTRUCTION PIPELINE

## B.1  DATA PREPARATION

We utilize data from AudioCaps (Kim et al., 2019), WavCaps (Mei et al., 2024), FSD50K (Fonseca et al., 2022), ESC50 (Piczak, 2015) and VGG-Sound (Chen et al., 2020) as raw data sources to construct our dataset. We first filter out samples with captions describing multiple sound sources to ensure each audio clip contains only a single sound event. We then apply active detection on each audio and obtain 10-second active segments. If the active part is shorter than 1 second, it is discarded, as short clips may lack sufficient information and be difficult to simulate as a moving source. If the audio duration is less than 10 seconds, we pad it to reach 10 seconds. Then, following Xu et al. (2024); Xie et al. (2024a); Huang et al. (2023b), a CLAP model (Wu et al., 2023) evaluates the similarity between each audio clip and its caption, discarding clips with similarity scores below 0.3. When an audio-caption pair is used in the simulation, we randomly select one from all the corresponding 10-second segments. Tab. B9 provides statistics for the raw data collected from different sources before and after processing.

Table B9: Statistic of raw data and processed data. Since some datasets contain long-form audio with a single global caption that may not represent the local description, activity detection and CLAP filtering with cropping is needed. So, the necessary decrease of data is acceptable.

| Data source | Before Processing | | After Processing | |
|---|---|---|---|---|
| | Num. of Audio | Total duration | Num. of Audio | Total duration |
| AudioCaps | 39597 | 110 Hours | 12,400 | 34.4 Hours |
| WavCaps: FreeSound | 262,300 | 6264 Hours | 10,844 | 30.4 Hours |
| WavCaps: BBC Sound Effects | 31,201 | 997 Hours | 4,047 | 11.2 Hours |
| WavCaps: SoundBible | 1,232 | 4 Hours | 683 | 1.9 Hours |
| WavCaps: AudioSet SL | 108,317 | 300 Hours | 8,294 | 23.0 Hours |
| FSD50K | 51,197 | 108.6 Hours | 16,981 | 47.2 Hours |
| VGGSound | 199,467 | 550 Hours | 157,230 | 436.8 Hours |
| ESC50 | 2,000 | 2.8 Hours | 2,000 | 2.8 Hours |

Table B10: The attributes frequently used to describe the simulated audio and corresponding options lists. Beyond these words, with the help of fine guidance and GPT, we still are able to generate a description of "*A dog is barking at 15°to the front left*". The moving speed ratio is used as the denominator, so faster speeds have smaller values.

| Attribute | Options List | Value |
|---|---|---|
| Scene size $r$ | outdoors | $100m$ |
| | large | $40 \sim 90m$ |
| | moderate | $20 \sim 40m$ |
| | small | $5 \sim 20m$ |
| Source direction $\theta$ | left | $N(180°, 121°)$ |
| | front left | $N(145°, 121°)$ |
| | front | $N(90°, 121°)$ |
| | front right | $N(45°, 121°)$ |
| | right | $N(0°, 121°)$ |
| Source distance (ratio) $d$ | far | $U(0.6, 0.9)$ |
| | moderate | $U(0.3, 0.6)$ |
| | near | $U(0.1, 0.3)$ |
| Movement | still | - |
| | moving | - |
| Moving speed (ratio) $\alpha$ | slow | $U(0.75, 0.85)$ |
| | moderate | $U(0.45, 0.55)$ |
| | fast | $U(0.25, 0.35)$ |
| | instantly | - |

As for the real-world subset (RW-world), we manually select around 200 samples from FairPlay (Gao & Grauman, 2019) for musical instruments and STARSS23 (Shimada et al., 2024) and SimBinaural (Garg et al., 2023) for audio events. Then, invite experts to write the descriptions of spatial audio.

## B.2 GPT-BASED ATTRIBUTES DERIVATION AND CAPTION TRANSFORMATION

**Attributes.** As mentioned previously, the audio simulator requires certain essential attributes to simulate realistic audio, including sound objects, scene size, sound source location, moving direction, and speed. The inferred attributes with corresponding audio clips are used to construct an audio-object pool and retrieve audio from images. Afterward, we convert the attributes selected by GPT-4 and GPT-4o into numerical values for audio simulation. All attributes and their mapping to values can be found in Tab. B10. The use of these attributes is detailed in Sec. B.4.

To obtain reasonable attributes and captions, we apply the Chain of Thought Prompting (CoT) (Wei et al., 2023) to enhance the common sense reasoning ability of GPT-4 and GPT-4o. We predefined several candidates to describe each attribute, with each being an attribute element, such as "far", or "near" for the distance attribute. We require GPT to select attribute labels that match the input information. For direction, we require GPT to provide more detailed values, such as when specific angles appear in the caption or when reasoning with images.

For attributes in audio-image pairs, we provide image-caption pairs with object positions to GPT-4o, and it infers all attributes because the image and text caption carry enough information. Based on the provided positions, GPT-4o can infer more precise object directions instead of selecting from a list. This makes our generation more accurate. However, for audio-caption pairs, we provide the caption to GPT-4, and it only infers room size and moving speed based on the caption, as other attributes are not related to the text caption. The other attributes of audio-caption pairs are randomly chosen from the attribute lists.

**Caption Transformation.** As part of the prompts, we also utilize GPT-4 and GPT-4o to transform raw captions into merged captions with positional and movement phrases. We also back up the merged captions without spatial and movement information for further training. Noticing that merging multiple audio-caption pairs results in long and complex captions, we instruct GPT to shorten the final output.

Appendix H presents all prompts used here with real examples.

### B.3 AUDIO RETRIEVAL FROM IMAGE

We retrieve audio from the audio-object pool mentioned earlier to construct image-audio pairs. The current visual-caption pair (Wu et al., 2023) widely used in VLM excels at conveying visual information but lacks plausible acoustic descriptions. For instance, when describing a photo of a woman, typical VLM descriptions focus primarily on the visual semantics, including behavior, appearance, and the surrounding environment, to form the description like "*A woman with the white shirt is standing on the left.*". However, if we aim to use language-driven approaches to generate corresponding stereo sounds for images, we seek to capture acoustic-rich details about the direction and sounds from the image, e.g., "*A woman's laughter comes from left side*". Therefore, we utilize LLM to obtain multiple alternative acoustic captions about objects in the image that may produce sound and finally obtain the description of the direction and possible moving of those sounds.

Since we choose a text modality with abstract semantics as a bridge to connect multi-channel audio and other modalities, we develop a method to obtain tri-modal triplets based on three modalities in order to further promote the development of multi-modal guided audio generation.

We employ two methods to utilize images as cues for audio retrieval: 1) 2-C Audio Retrieval via Language Bridge: By using sentence embedding, we match visual and audio elements through text, considering samples with a threshold exceeding 0.9 to be similar. 2) 1-C Audio Retrieval with Simulation: Building on previous research, we initially identified the sound subjects from the GPT-4o. Utilizing these subjects, we extract multiple samples from the audio-subject pool. Taking a holistic view, these two methods are basically the same.

Finally, we allow each image to correspond to up to 10 possible audio clips, rather than just one. This approach of one-to-multi correspondence enhances the diversity of our I2A generation.

### B.4 AUDIO SIMULATION

In this section, we introduce the details of audio simulation and how we use the attributes. We use audio simulators like Pyroomacoustics and gpuRIR to simulate spatial audio.

For static audio, Pyroomacoustics uses inputs such as room size, microphone location, sound source locations, and RT60. RT60 represents the time it takes for sound energy to diminish by 60 dB once the source stops. We randomly sampled RT60 values between 0.3 and 0.6 seconds to ensure data diversity. The simulated rooms are cubic, with sizes defined as

$$[R_0, R_1, R_2] = [r + \xi_{r0}, r + \xi_{r1}, r + \xi_{r2}], \tag{9}$$

where $\xi_{ri} \sim U(-0.1r, 0.1r)$ and $r$ is the room size. The room size value can be obtained from its attribute label inferred from GPT-4 or GPT-4o. For the room size attribute, each label corresponds to a specific range. The value is randomly sampled from these ranges: ["small": $U(5, 20)$, "moderate": $U(20, 40)$, "large": $U(40, 90)$, "outdoor": 100]. An anechoic room mode is used for outdoor scenes. Microphones are positioned centrally, using a 2-microphone array with a 0.16–0.18m separation. The array's center is

$$[M_0, M_1, M_2] = [\frac{R_0}{2} + \xi_{m0}, \frac{R_1}{2} + \xi_{m1}, \frac{R_2}{2} + \xi_{m2}], \tag{10}$$

where $\xi_{mi} \sim U(-0.1r, 0.1r)$. The two microphones are positioned at $[M_0, M_1 \pm \xi_c, M_2]$, where $\xi_c \sim U(0.08, 0.09)$. We sample the angle $\theta$ from a normal distribution with a standard deviation of $11°$ based on the source direction attribute label: ["left": $N(180°, 121°)$, "front left": $N(135°, 121°)$, "directly front": $N(90°, 121°)$, "front right": $N(45°, 121°)$, "right": $N(0°, 121°)$]. The distance ratio $\alpha_d$ that controls

the distance between the microphone array center and the source is sampled from ["near": $U(0.1, 0.3)$, "moderate": $U(0.3, 0.6)$, "large": $U(0.6, 0.9)$]. The sound source is located at this angle, with a distance from the microphone array center of

$$d = \alpha_d \times \min(R_0 - M_0, R_1 - M_1, M_0, M_1). \tag{11}$$

Using variables in equation 10 and equation 11, the position of the source $\mu_{begin}$ can be calculated as

$$\mu_{begin} = [M_0 + d\sin\theta, M_1 + d\cos\theta, M_2]. \tag{12}$$

For dynamic scenes, gpuRIR simulates moving sources with specified trajectories. Room size, microphone location, and source position are determined as in static scenes. In the outdoor scene, the absorption weights of six walls are set to $1 \times 10^6$ to simulate an anechoic chamber. Additional parameters include the speed and endpoint of the moving source $\mu_{end}$. The speed of the moving source is

$$V = \frac{\delta_{total}}{T}, \tag{13}$$

where $\delta_{total}$ presents the displacement from the start position to the end position, and $T$ donates the moving interval. We can obtain $T$:

$$T = \alpha \times T_{total},$$

where $\alpha$ is sampled based on moving speed attributes: ["slow": $U(0.75, 0.85)$, "moderate": $U(0.45, 0.55)$, "fast": $U(0.25, 0.35)$]. We also sample a $T_{begin} = U(0, 0.15) \times T_{total}$ to indicate when the source starts to move. GpuRIR can simulate the moving source by the source trajectory $[P_0, P_1, ..., P_t]$, and each point on the trajectory $P_t$ are calculated as

$$P_t = \begin{cases} \mu_{begin} & \text{for } t < T_{begin} \\ \mu_{begin} + tV & \text{for } T_{begin} \leq t \leq T_{begin} + T, \end{cases} \tag{14}$$

where $\mu_{begin}$ is the start position of source. We calculate the source position every 10ms. In instantly moving scenes, the source changes position to end position suddenly at a random $t_{move} = U(0.2, 0.8) \times T_{total}$.

In the mixed subset, we use gpuRIR to simulate both stationary and moving sources in an outdoor scene. All other operations are the same as in the dynamic scene.

In the dataset construction process, some attribute labels are randomly chosen from the list for audio-caption pairs, like source direction, distance, and moving speed. We record all attribute values (including RT60 and microphone location) for model training and statistics. During inference, all attribute labels can be inferred from input captions or images with object positions.

## C  DATASET AND BENCHMARK STATISTICS

### C.1  AUDIO-CAPTION SUBSET

The dataset is divided into **five** subsets based on the number and motion state of the sound source:

- Single Static Subset (SS-set): One stationary audio.
- Double Static Set (DS-set): Two stationary sound sources.
- Single Dynamic Set (SD-set): A single moving sound source.
- Mixed Set (M-set): 1 to 4 sound sources including both stationary and moving sources
- Real-world Set (RW-set): naturally recorded audios with manually written descriptions for testing.

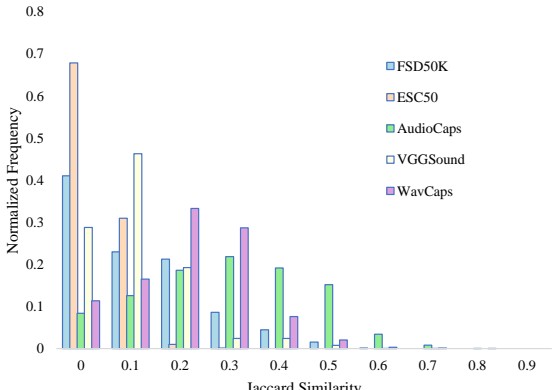

Figure C5: Jaccard similarity between raw descriptions and our rewritten captions in the single static subset. We follow Mei et al. (2024) to conduct this analysis to show a generally low level of lexical overlap across various sources.

For the single static subset, we utilize all single-source clips. To synthesize an audio clip in the double static subset, we sample and mix two different audios from the single-event database. In the single dynamic subset, sound sources move in two modes: gradual, where the source moves from the start to the end position at a constant speed; and instant, where the source changes position suddenly. For "instant", we perceive in the caption that two sound sources emit sound sequentially from different positions. For the mixed subset, we choose 1 to 4 sources from the single-event database, with the possibility of moving. All captions are rewritten by LLM to include spatial and moving instruction. The test and validation subsets from AudioCaps are used to construct our test set, while all other data is used for the training set. The specifics of our dataset are detailed in Tab. C11. As mentioned in Appendix B.2, we also retrieve audio from images to construct image-audio captions. Tab. C12 shows the detailed number of each modality pair in BEWO-1M.

Table C11: Audio-caption subsets statistics of BEWO-1M dataset. The four main subsets have rather uniform distribution. Moreover, a real-world subset is constructed manually to verify the perception consistency between simulation and real-world.

| Subset | Total Duration | Num. of Audio | Avg. Caption Len | Max. Caption Len | Min. Caption Len |
|---|---|---|---|---|---|
| Single Static Subset | 835.03 Hours | 319,259 | 12.44 Words | 52 Words | 2 Words |
| Double Static Subset | 573.95 Hours | 205,880 | 19.54 Words | 92 Words | 4 Words |
| Single Dynamic Subset | 572.15 Hours | 205,975 | 13.06 Words | 77 Words | 3 Words |
| Mixed Subset | 791.74 Hours | 285,028 | 24.20 Words | 64 Words | 6 Words |
| Real-World Subset | 0.6 Hours | 200 | 14.17 Words | 34 Words | 6 Words |

Table C12: The details of modality pairs involved in BEWO-1M.

| BEWO-1M | number |
|---|---|
| Text-Audio pairs | 1,016k |
| Image-Text pairs | 3.2k |
| Image-Audio triplets | 113K (20K unique audios) |

Fig. C5 presents the Jaccard similarity scores, it shows that our caption is quite different from the original caption. Coupled with the observed increase in length that includes spatial information, this suggests that transformed captions have significantly changed from the original descriptions and incorporate substantial additional information.

Table C13: Examples of captions in our dataset. The final caption should not only involve the spatial context but also maintain the concise of the sentence.

| Task | Raw Caption | Final Caption |
|---|---|---|
| Single still | A cell phone is vibrating. | A cell phone is vibrating on the right side of the scene. |
| Double still | Printer printing. Playing didgeridoo. | The printer is printing on the right of the scene, while the person is playing the didgeridoo directly in front. |
| Single dynamic | Trumpet is being played. | Trumpet sound moves from right to front left at a moderate speed. |
| Mixed | A vehicle's engine starts to die down. Young children are whistling and laughing. | An engine slowly dying down is noticed on the left, as children's laughter and whistling gently move from directly in front to the left. |

Table C14: Correctness statistics of captions for each training subset without manual correction. A random samples from each subset was checked to ensure captions match the simulated audio.

| Subset | Correctness |
|---|---|
| Single Static Subset | 97.87% |
| Double Static Subset | 87.23% |
| Single Dynamic Subset | 95.74% |
| Mixed Subset | 91.48% |

We show some caption examples in Tab. C13, more examples are provided in our demo page https://immersive-audio.github.io/. We offer the raw caption and final caption to the readers for reference. Tab. C14 presents the correctness for different training subsets without manual check. We also test the overall accuracy of attribute inference and caption transformation in the inference process, and overall, and it achieves an acceptable performance of 91.52%.

## C.2    I2A BENCHMARK SUBSET

As mentioned in Appendix B.3, we retrieve audio based on the image-caption pairs to construct our dataset. We use COCO-2017 (Lin et al., 2014) to obtain the train set and test set of I2A. The LLM we use to obtain the acoustic semantic description is GPT-4o. During text retrieval, we extract the sentence embedding using SentenceTransformer[4]. In constructing Image-Text-Audio triplets, we employ the FAISS [5] library to perform exact retrieval using Euclidean distance (L2). Our algorithm restricts each image to match with up to 10 audio files. For the approach of 1-C retrieval with simulation, the extra simulation based on inferred attributes similar to Appendix B.4 should be carried out after retrieval. Ultimately, we generate 113k triplets to form a dataset comprising 3.2k images and 20k audios. For the test set, experts are invited to evaluate the audio and image correspondence and drop all the low-quality samples manually.

## C.3    INTERACTIVE2A BENCHMARK SUBSET

We select 150 images from the COCO-2017 test set and use them to allow real users to choose the objects of interest. Each image is annotated by at least 4 participants, who identify the objects using boxes and points using makesense[6]. In total, there are about 300 boxes and 300 points used for testing.

## C.4    DATASET LICENCE

We have taken investigation into each previous dataset involved in our BEWO-1M. For the purpose of open access, we follow each dataset involved in BEWO-1M and apply the license including the Creative Commons Attribution (CC BY) license to any Author Accepted Manuscript version arising.

---

[4]https://sbert.net/

[5]https://github.com/facebookresearch/faiss

[6]https://www.makesense.ai/

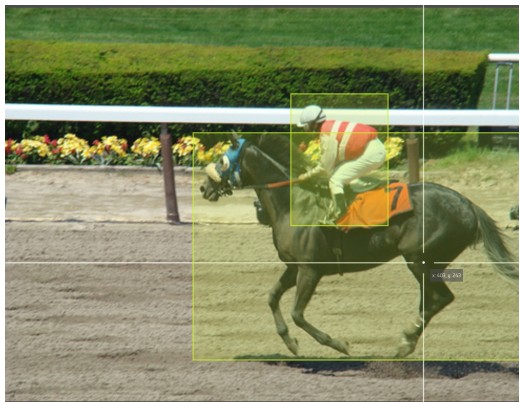

Figure C6: Interactive box labeling.

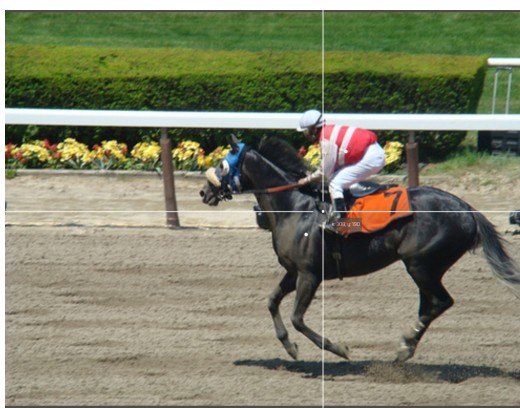

Figure C7: Interactive point labeling.

# D    BASELINE DETAILS

## D.1    TEXT-TO-SPATIAL-AUDIO BASELINE

We select three popular T2A models to construct the baselines: AudioLDM 2, Make-an-audio 2, and Stable Audio Open. The model structures of them remain unchanged. They extract features from the text using T5 and CLIP and use these features to guide the audio generation process. AudioLDM 2 and Make-an-audio 2 can only generate mono audio, so we apply a learnable conditional filter (Zhou et al., 2020) based on U-Net, which has been proven effective. Same as the previous approach (Xu et al., 2021; Zhou et al., 2020), this filter also leverages position features as conditions to derive a suitable mask based on ITD and ILD training in a mono-to-binarual reconstruction task. This mask, combined with basic operations on the generated mono audio, results in stereo audio. To facilitate a more comprehensive comparison of different baselines, we fine-tune and optimize the previously advanced mono audio generation models while ensuring consistency in all training datasets. Stable Audio Open can generate stereo audio directly, so we directly fine-tune it based on our training dataset.

While the filter performance appeared promising according to Xu et al. (2021); Zhou et al. (2020), when confronted with our large-scale dataset, low performance persisted despite our efforts to scale up the number of layers and parameters, as illustrated in Tab. 3. Moreover, on the demo page, we have presented qualitative comparative cases showing that techniques relying on mono-channel generation models and filters exhibit fluctuating ILD and inadequately capture ITD. This underscores the strengths of our end-to-end dual-channel audio generation approach.

## D.2    IMAGE-TO-SPATIAL-AUDIO BASELINE

Similar to the construction method of the Text-to-Spatial-Audio baseline, we utilize an Open-domain Image-to-Audio model (Xing et al., 2024) to generate monophonic audio. Additionally, we sequentially append a filter based on image conditions at the end of the model to facilitate the transition from mono-channel to binaural audio.

### D.3 INTERACTIVE-TO-SPATIAL-AUDIO BASELINE

See2Sound (Dagli et al., 2024) also leverages the RoI strategy to capture positional information of different objects. See2Sound begins by performing universal image segmentation to identify regions of interest for various objects in the image. Consequently, we naturally employ a filtering and selection approach to extract a small number of RoIs in an image based on user-interacted bounding boxes and points, while keeping other components like the Depth-anything model unchanged. This zero-shot method is then used to establish the baseline.

Additionally, to create other comparable baselines, we enhance the Image-to-Spatial-Audio model (Xing et al., 2024) by incorporating the SAM to set the relevant pixels transparent. By focusing on the region of interest that is not transparent, we then proceed with spatial audio generation based on the processed image as discussed in Sec. D.2.

## E MODEL DETAILS

### E.1 MODEL STRUCTURE

**Continuous Auto-encoder:** We use the oobleck (Jang et al., 2023) pre-trained by Stable[7]. The configuration of our model includes an audio setup with 2 channels. The encoder features a hidden size of 128, while the decoder operates with an input channel size of 64 and an overall channel configuration of 128. The architecture employs channel multiples set at [1, 2, 4, 8, 16] and utilizes downsampling ratios of [2, 4, 4, 8, 8], effectively capturing the complexities of the audio data. This configuration is crucial for achieving the desired fidelity and efficiency in our audio processing tasks. It is trained by multi-resolution STFT loss with the left, right, summation, and subtraction of the dual channel. Moreover, a discriminator like EnCodec (Défossez et al., 2022) can refine audio with multi-resolution to achieve finer quality.

**Diffusion Transformer:** Our diffusion model is tailored to process 10-second audio samples with two channels upon Stable[8]. It integrates a conditioning mechanism consisting of multiple configurations: a text-based prompt processed by a T5 transformer model ("T5-base" with a maximum length of 128), and azimuth state encoding with an output dimension of 768. The conditioning dimension is set at 768. The diffusion component utilizes a Diffusion transformer (DiT) with settings that include 64 input/output channels, an embedding dimension of 1536, 24 layers, 24 attention heads, and both local and global conditioning dimensions of 768 and 1536, respectively. Notably, the transformer operates with projecting condition tokens and adheres to a continuous transformer architecture. For training, an exponential moving average (EMA) is used alongside an AdamW optimizer with a learning rate of $2e-5$, beta values of [0.9, 0.999], and a weight decay of $1e-3$, complemented by an InverseLR scheduler that features an inv_gamma of $1e6$, a power of 0.5, and a high warmup proportion of 0.99. This configuration underscores our commitment to refining audio quality and temporal alignment in generative tasks. During inference, we use the DPMSolver++ Lu et al. (2022) for 100 steps with classifier-free guidance (scale of 6.0). The adapter mentioned in this paper is a one-layer MLP.

**Text Encoder:** A pre-trained T5-base[9] encoder is utilized in the main experiment. The CLAP encoder we use in the ablation study is from Laion[10].

**Image Encoder:** The classic Mask-RCNN (He et al., 2017) is used as the detection model. Since the regional feature of Mask-RCNN is about the class itself rather than the behavior, the regional CLIP feature is then used to understand both the behavior and class of each object. Our bidirectional multimodal encoder

---

[7]https://github.com/Stability-AI/stable-audio-tools

[8]https://github.com/Stability-AI/stable-audio-tools

[9]https://huggingface.co/google-t5/t5-base

[10]https://huggingface.co/laion/clap-htsat-unfused

comprises a stack of 12 transformer blocks, each featuring a self-attention layer and a fully connected layer, enhanced by residual connections. Similarly, our decoder consists of another 12 blocks stack mirroring the encoder's architecture but with each block augmented by an additional cross-attention layer. The text prefix is set to "*image acoustic captioning:*". The image network with regional perception is first trained as the image caption task and supervised by the acoustic description of the image. We train this bidirectional multimodal encoder for 100 epochs on $1 \times$ NVIDIA RTX 4090. After training the T2A model, the pertained image encoder is built on top of the T2A model. After another 100 epochs of fine-tuning, all the I2A tests and Interactive2A tests are carried out on the same checkpoints.

**Azimuth Fusion Module:** This module consists of a 4-layer of cross-attention module with 4 heads. The $m$ mentioned in Fig. 4 equals to 4.

**Interactive Matching:** 1) A straightforward method for matching and filtering is at the coordinate level. Initially, we use SAM to generate a high-quality mask, which is then matched with region coordinates $\mathcal{C}$ detected globally using the Intersection over Union (IoU). To enhance this baseline approach, we adopt the maximum continuation strategy, enabling the selection of multiple areas of interest for feature generation. 2) We further advance this by introducing a feature-level matching technique, where we conduct feature-level retrieval using CLIP between interactive areas and detected regions.

## E.2 Dimension Details

$L$ is variable depending on the complexity of text and image. $L_s$ is the sample size that relates to the sampling rate and audio interval. But the overall condition we use to condition the whole diffusion is 768-channel, which means $L_{\text{reg}} = d_{text} = d_{img} = L_{clip} = d_{\text{time}} = 768$. The variance $\sigma = 4$ is obtained empirically from the real-world distribution. The maximum region number is $N_{max} = 38$ following Cho et al. (2021). $L_{\text{azi}}$ is set to 64 and $L_{\text{cls}}$ is set to 80.

## E.3 Training matrix and caption composition

During data construction, in addition to the spatial caption, we also transform and preserve the caption without spatial phrases for future training enhancement.

During T2A model training, three types of guidance compositions are used: coarse azimuth matrix with the transformed caption, fine azimuth matrix with the original caption, and fine azimuth matrix with the transformed caption. This joint training method requires the model to consider both the description and azimuth state matrix to generate the spatial audio.

However, for I2A training, only the fine azimuth matrix is used to train the model.

## F Metrics details

### F.1 1-C Metric Details

We use several common metrics in 1-C evaluation. 1) Fréchet Distance (FD) measures the similarity between two distributions. 2) Inception Score (IS) evaluates data generation by assessing diversity and resemblance to real data. 3) Kullback-Leibler divergence (KL) quantifies divergence between two probability distributions. 4) Fréchet Audio Distance (FAD) assesses audio quality by comparing generated audio to real audio samples. 5) CLAP score evaluates the alignment between audio and text for coherence in multimedia tasks. 6) overall impression (OVL) subjectively measures the general quality and appeal of generated content. 7) Audio-text relation (REL) assesses coherence and relevance between audio content and accompanying text. 8) The CLIP score measures the alignment between images and text using the CLIP model's shared space.

### F.2 GCC-PHAT AND STEREOCRW DETAILS

The objective of the interaural time difference estimation problem is to ascertain the difference in arrival times of a sound at two microphones. In a stereo recording scenario with $x_1, x_2 \in \mathbb{R}^n$ denoting waveforms, and a function $\mathbf{h} : \mathbb{R}^n \to \mathbb{R}^{n \times d}$ which calculates features for each temporal sample, a frequently employed approach involves selecting a time delay $\tau$ that maximizes the generalized cross-correlation.

$$R_{\mathbf{x}_1,\mathbf{x}_2}(\tau) = \mathbb{E}_t \left[ \mathbf{h}_1(t) \cdot \mathbf{h}_2(t - \tau) \right] \tag{15}$$

where $h_i = h(x_i)$ are the features for $x_i$, and $h_i(t)$ is the $d$-dimensional feature embedding for time $t$. The visualizations of TDOA over the I2A task can be found in Fig. F8.

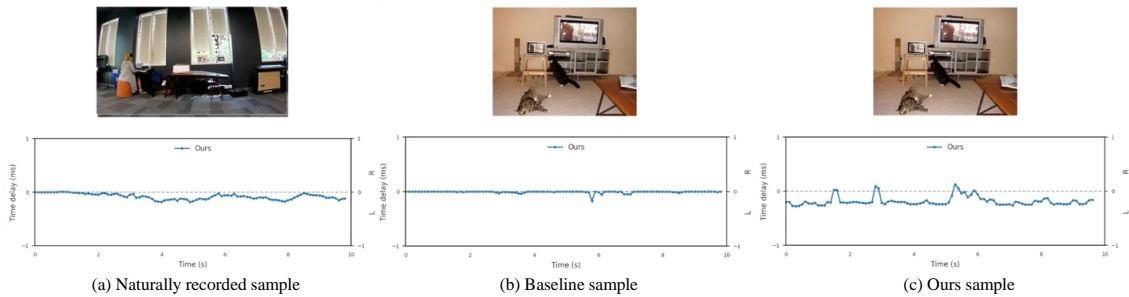

(a) Naturally recorded sample      (b) Baseline sample      (c) Ours sample

Figure F8: The visualization of DTOA across the time. (a) The real-world recorded sample is selected from FairPlay. (b) Baseline sample from BEWO-1M image test set. (c) Our sample BEWO-1M from our image test set.

**GCC-PHAT**: The popular Generalized Cross Correlation with Phase Transform (GCC-PHAT) technique (Knapp & Carter, 1976; Salvati et al., 2021) whitens the audio signal by normalizing it with the magnitude of the cross-power spectral density. This method offers the maximum likelihood solution in specific ideal, low-noise scenarios.

**StereoCRW**: As a more advanced deep learning-based method for ITD estimation, StereoCRW (Chen et al., 2022b) aims to autonomously acquire these connections by utilizing advancements in tracking techniques. By modifying the contrastive random walk approach, a cycle-consistent representation is developed from unlabeled stereo sounds.

**GCC MSE** and **CRW MSE**: This metric computation process involves averaging the ITD results of the entire audio and then calculating the mean square error, treating the video as a perceived difference in overall auditory channels. While this method exhibits high accuracy and computational efficiency for determining the direction of a single sound source, its limitations are apparent in scenarios involving moving sources and multiple sources in different directions. Specifically, when calculating ITD, a time unit of 0.1 seconds is employed. Subsequently, the overall video's ITD is augmented for significance by multiplying it with a coefficient of 100.

**FSAD**: To address scenarios with moving and multiple sound sources, we adapted the FAD methodology by leveraging the features from the final layer of StereoCRW, supplemented with adaptive pooling to derive a 2560-dimensional representation for the entire audio. Subsequently, FD is computed using this representation. Through this approach, we have successfully developed a model for evaluating the generation task in complex sound source scenarios.

All evaluation codes will be publicly accessible.

### F.3 FSAD ANALYSIS

FSAD processes the ITD features from multiple time slots to obtain the distance of temporal perception. Given that the GCC MSE and CRW MSE represent only the temporal average ITD, in some moving scenarios, the effects from the left and right parts counteract each other. In the samples of Table.F15, changes in the moving direction between No.1 and No.2 are not reflected by the MSEs, but perfectly reflected by the FSAD. Therefore, in moving scenarios, FSAD can be treated as a better metric.

Table F15: Some cases to show the effectiveness of FSAD.

| Type | No. | Source audio | Target audio | GccPHAT MSE | CRW MSE | FSAD |
|------|-----|--------------|--------------|-------------|---------|------|
| Single Dynamic | 1 | A car is moving from left to right. | A car is moving from left to right. | 0.73 | 0.63 | 0.08 |
| Single Dynamic | 2 | A car is moving from left to right. | A car is moving from right to left. | 0.58 | 0.45 | 4.43 |
| Single Stationary | 3 | A duck is quacking on the left. | A duck is quacking on the right. | 132.13 | 107.19 | 4.90 |

### F.4 MOS DETAILS

Table F16: Objectives and Interfaces of different tasks.

| Prompt Type | Objective | Interface |
|-------------|-----------|-----------|
| Text (1-C) | Fidelity & Consistency (previously called OVL and REL) | Figure F9 |
| Image (1-C) | Fidelity & Consistency (previously called Fidelity and Relevance) | Figure F10 |
| Bounding Box | Event relevance & Direction relevance | Figure F11 |
| Point | Event relevance & Direction relevance | Figure F12 |
| Image | Event relevance & Direction relevance | Figure F13 |
| Text | Event relevance & Direction relevance | Figure F14 |

We conduct all subjective evaluations online with 15 participants using Amazon Mechanical Turk[11] (AMT) (Crowston, 2012). Listeners evaluate the fidelity, consistency, and relevance of the events or direction of each sample on a 5-point Likert scale while listening through headphones in a quiet setting. The evaluation consists of six tasks:

- Participants are provided with an audio clip and an optional text caption. They evaluate the naturalness of the audio and its relevance to the caption. (See Fig. F9).

- Participants receive an audio clip and an optional image. They rate the naturalness of the audio and its relevance to the image. (See Fig. F10).

- Participants are presented with an image containing a bounding box and corresponding audio. They score the agreement between the sound event and the object in the bounding box, as well as the directional accuracy of the sound relative to the bounding box. (See Fig. F11).

- Participants receive an image with a point and corresponding audio. They evaluate how well the sound event matches the pointed object and the directional accuracy of the sound relative to the point. (See Fig. F12).

- Participants are given an entire image and corresponding audio. They assess the alignment between the sound event and the image, and how accurately the sound direction matches the image. (See Fig. F13).

- Participants receive a caption and corresponding audio. They score the alignment between the sound event and the caption, and the directional accuracy of the sound concerning the caption. (See Fig. F14).

---

[11]https://requester.mturk.com/

The evaluation objectives and interfaces are shown in Tab. F16. In the first four tasks, the scale options are: "1. Excellent - Completely faithful events/direction", "2. Good - Mostly faithful events/direction", "3. Fair - Equally faithful and inconsistent events/direction", "4. Poor - mostly inconsistent events/direction", "5. Bad - Completely inconsistent events/direction". In the other two tasks, the scale options are: "1. Excellent - Completely natural/faithful", "2. Good - Mostly natural/faithful", "3. Fair - Equally natural/faithful and unnatural/inconsistent", "4. Poor - mostly unnatural/inconsistent", "5. Bad - Completely unnatural/inconsistent". Note that the scoring order in the Turk interface is reversed compared to that in our paper: in Turk, a score of 1 represents the best, while in the paper, a score of 5 represents the best.

## G  SUPPLEMENTARY EXPERIMENTS

### G.1  COMPARISON OF 1-C METRICS ON CLOTHO

Although it is now more common to use real-data AudioCaps as the test set including Make-An-Audio 1&2, AudioLDM 1&2, Tango 1&2, we still manage to conduct zero-shot experiments on Clotho (Drossos et al., 2020) to showcase the comparative performance. As shown in Table. G17, our SpatialSonic demonstrates comparative performance across almost several metrics, particularly in FD, FAD, and ISc. It slightly lags behind the popular methods on some metrics but overall, it appears to be the effective model among those listed for generating high-quality and relevant audio content.

Table G17: The 1-C zero-shot generation comparison on less usual dataset Clotho. All the methods listed below is tested on the evaluation set of Clotho.

| Model | Objective | | | | | Subjective | |
|---|---|---|---|---|---|---|---|
| | CLAP↑ | FD↓ | FAD↓ | ISc↑ | KL↓ | OVL↑ | REL↑ |
| Make-an-audio | 0.331 | 27.32 | 6.10 | 6.94 | 3.15 | 3.27 | 3.32 |
| Make-an-audio2 | 0.343 | 19.10 | 3.48 | 8.19 | 2.47 | 3.56 | 3.58 |
| Audioldm2 | 0.340 | 25.39 | 3.49 | 7.93 | 2.62 | 3.47 | 3.48 |
| Tango2 | **0.363** | 22.72 | 3.39 | 9.66 | **2.21** | 3.53 | 3.49 |
| SpatialSonic (Ours) | 0.361 | **18.81** | **3.37** | **10.31** | 2.36 | **3.61** | **3.63** |

### G.2  COMPARISON OF 1-C METRICS ON DIFFERENT SUBSETS

Previous mono-channel metrics tested on dual-channel text-to-audio systems have revealed significant limitations. The CLAP score struggles to adapt to the challenges posed by dual-channel audio and longer texts. Since CLAP is trained on mono-channel data and text lacking directional information, its utility in evaluating dual-channel audio is insufficient. Metrics such as FD, IS, KL, and FAD, though widely used, are originally designed for mono-channel applications and do not fully capture the quality of channel differences in dual-channel generation. Therefore, the overall mono-channel evaluation remains somewhat inadequate for stereo audio.

For deeper insights, we analyze mono-channel evaluation metrics across various subsets in BEWO-1M by channel-wise average in Tab. G18. Notably, CLAP's overall evaluation scores are relatively low due to its incapacity to handle long-form captions and spatial phrases. Furthermore, from the mono-channel evaluation metrics, it is evident that audio generation for single sources is relatively straightforward. However, as complexity increases with factors like long captions, multiple sources, and motions, the difficulty of generation also rises, leading to a corresponding variance in quality.

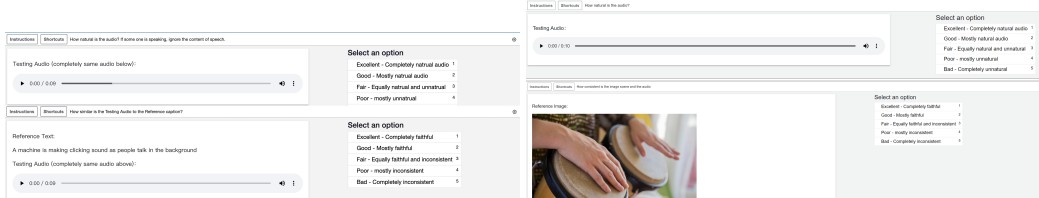

Figure F9: MOS interface of audio-text pair.    Figure F10: MOS interface of audio-image pair.

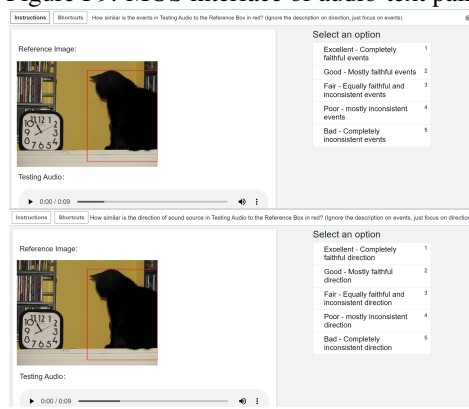

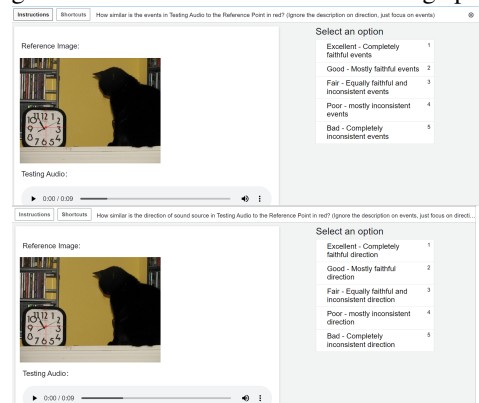

Figure F11: MOS interface of bound box.    Figure F12: MOS interface of point.

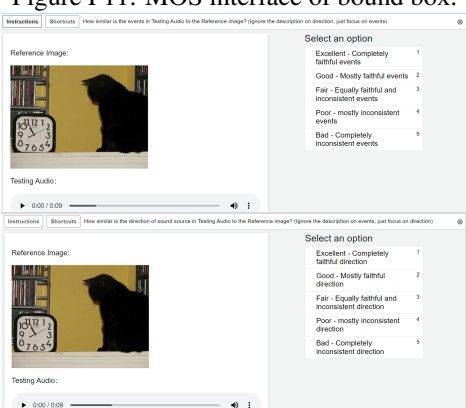

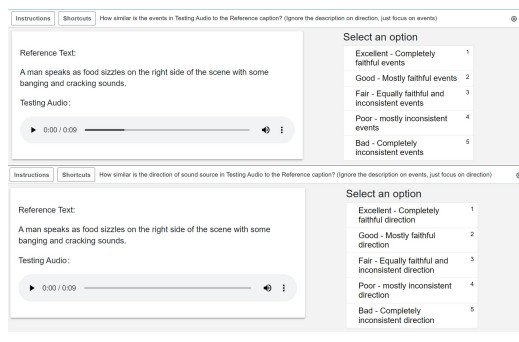

Figure F13: MOS interface of the entire image.    Figure F14: MOS interface of text caption.

Table G18: 1-C audio objective comparison of different subsets in our dataset. RW-set is a completely unseen set, so there still exists a distribution gap between ground truth audio and generated audio. But it does not impediment the validation of using such a matrix on source azimuth evaluation.

| BEWO-1M | Objective | | | | |
|---|---|---|---|---|---|
| Subset | FD↓ | IS↑ | KL↓ | FAD↓ | CLAP↑ |
| SS-set | 24.62 | 10.33 | 1.66 | 2.30 | 0.4026 |
| SD-set | 17.33 | 13.09 | 1.60 | 2.14 | 0.3501 |
| DS-set | 25.20 | 7.74 | 2.09 | 2.16 | 0.3437 |
| M-set | 21.77 | 9.59 | 2.25 | 1.79 | 0.3824 |
| RW-set | 33.37 | 5.87 | 1.36 | 4.26 | 0.3873 |

## G.3 COMPARISON ON PREVIOUS I2A BASELINE

ImageHear (Sheffer & Adi, 2023) is an evaluation dataset containing images of specific categories used in previous mono-channel audio. Given that our model relies on a close-set detection model on COCO-2017 for region-level perception, certain objects in ImageHear are undetectable. To address this limitation, we employ the GPT-4o and a suitable prompt to generate a dual-channel output in a language-driven approach, which is then averaged to produce a mono-channel video. Our evaluation of performance metrics is outlined in Table. G19.

Table G19: Image guided single channel audio generation test on Imagehear. V2A-mapper[†] is not open-source for now, in which we use the demo in their demo page to carry out the testing.

| Method | Objective | Subjective | |
|---|---|---|---|
| | CLIP-Score | Fidelity | Relevance |
| Im2Wav | 9.843 | 3.43 | 3.13 |
| CLIPSonic-IQ | 11.392 | 3.70 | 3.03 |
| V2A-mapper[†] | 11.950 | **3.77** | 3.41 |
| SpatialSonic(Ours) | **11.976** | 3.75 | **3.42** |

Overall, our model demonstrates commendable generalization on the ImageHear dataset in comparison to mono-channel audio.

Furthermore, based on the previous work Dagli et al. (2024), we also attempt to compute previous evaluation metrics. We utilize the audio outputs and images to create modified scene-guided audio generated using AViTAR. We present the average audio similarity scores among multiple such generated audios in Table. G20.

Table G20: Image guided multi-channel audio generation on AViTAR.

| (+AViTAR) | See2Sound Eval Set | | | | BAG Bench Eval Set | | | |
|---|---|---|---|---|---|---|---|---|
| | MFCC-DTW↓ | ZCR↑ | Chroma↑ | Spect↑ | MFCC-DTW↓ | ZCR↑ | Chroma↑ | Spect↑ |
| CoDi | 0.800 $\times10^{-3}$ | 0.80 | 0.70 | 0.85 | 0.730 $\times10^{-3}$ | 0.78 | 0.53 | 0.37 |
| SEE-2-SOUND | 0.034 $\times10^{-3}$ | 0.95 | 0.77 | 0.95 | 0.026 $\times10^{-3}$ | 0.91 | 0.61 | 0.51 |
| SpatialSonic(Ours) | **0.027** $\times10^{-3}$ | **0.97** | **0.77** | **0.97** | **0.021** $\times10^{-3}$ | **0.93** | **0.63** | **0.53** |

## G.4 $\chi^2$-TEST OF USER PREFERENCE

In previous studies (Dagli et al., 2024), researchers initially assume a uniform distribution of generated audio quality and spatial information. Then they survey different samples and utilize chi-square ($\chi^2$) tests to reject this hypothesis of uniform distribution. This process further substantiates a strong correlation between the generated spatial audio and human perception elements like audio quality and spatial awareness. While this evaluation metric holds statistical validity, it proves insufficient for supervised generation assessment.

Nevertheless, we still conduct statistical analysis of the survey, yielding a series of significant $p$-values in Tab. G21. We maintain the initial hypothesis that the responses adhere to a uniform distribution. Instead of

relying on continuous perception as in (Dagli et al., 2024), we formulate multiple-choice questions for each target. In these questions, we designate 3, 4, and 5 as correct choices on a 1-5 scale. Our user testing involves approximately 120 human evaluators, and we present the relevant metrics from our human evaluation. This $p$-value indicates that users express a statistically significant preference for spatial audio quality, sound direction, and overall audio quality.

Table G21: $\chi^2$-test with the hypothesis of the responses following a uniform distribution. The significance of human preference is evaluated with $\chi^2$-tests at $p < 0.05$.

| Metric | $p$-value ($\downarrow$) |
|---|---|
| **Spatial Audio Quality** | |
| Realism | $3.99 \times 10^{-4}$ |
| Immersion | $5.18 \times 10^{-3}$ |
| Accuracy | $7.97 \times 10^{-6}$ |
| Clarity | $9.08 \times 10^{-3}$ |
| Consistency | $2.09 \times 10^{-6}$ |
| **Audio Identification** | |
| Overall Localization | $6.52 \times 10^{-8}$ |
| Audio Direction Identification | $2.27 \times 10^{-8}$ |
| Distance Identification | $1.93 \times 10^{-3}$ |
| **Audio-Image Matching** | |
| Events Identification | $2.09 \times 10^{-6}$ |
| Spatial Identification | $1.81 \times 10^{-7}$ |
| **Audio-Bounding Box Matching** | |
| Events Identification | $1.26 \times 10^{-6}$ |
| Spatial Identification | $3.18 \times 10^{-6}$ |
| **Audio-point Matching** | |
| Events Identification | $9.11 \times 10^{-6}$ |
| Spatial Identification | $1.82 \times 1.5^{-5}$ |

## G.5 AUDIO-LANGUAGE RETRIEVAL

We follow the Audio-Language Retrieval experiments of WavCaps (Mei et al., 2024), where the retrieval model learns Acoustic Semantic Embeddings (ASE) to map audio clips closer to their paired captions in the embedding space.

a) **Models**: Our model architecture integrates an audio encoder for audio representation and a language encoder for text representation. Specifically, we employ HTSAT (Chen et al., 2022a), a transformer network, as the audio encoder and a pre-trained BERT (Devlin et al., 2019) network as the text encoder. To project features into a shared embedding space, we implement a 2-layer MLP as the adapter for both encoders.

b) **Experimental Setup**: For single-channel experiments, we first establish baseline models trained on AudioCaps and Clotho datasets. We then develop zero-shot models using WavCaps and BEWO-1M, utilizing 30% of the total training data from BEWO-1M. For stereo audio in BEWO-1M dataset, we perform channel averaging to convert it to mono. To demonstrate BEWO-1M's pre-training capability, we subsequently fine-tune these zero-shot models using AudioCaps and Clotho datasets, respectively. The baseline and zero-shot models are trained for 15 epochs with a batch size of 128 and a learning rate of $5 \times 10^{-5}$ with the Adam optimizer while fine-tuning is conducted for 20 epochs on AudioCaps and Clotho. All audio inputs are standardized to 10-second segments through cropping or padding, and model checkpoints are selected based on validation performance. We evaluate the performance using Recall at rank $k$ (R@$k$) on the test sets of AudioCaps and Clotho. R@$k$ is 1 if the correct match appears in the top $k$ retrieved items, and 0 otherwise, averaged across all queries. For dual-channel experiments, we maintain settings identical to those of the single-channel setup. We first establish a baseline by evaluating the BEWO-1M test set on a WavCaps-pretrained model using channel averaging. Another dual-channel retrieval model is trained on 30% data of the BEWO-1M training set.

Table G22: The evaluation results of 1-C audio-language retrieval on the test sets of AudioCaps and Clotho. A higher score means better performance. "ZS" refers to zero-shot, "FT" refers to fine-tune, "A→B" means the model is pre-trained on the dataset "A" and then fine-tuned on the dataset "B".

| Model | Training Data | AudioCaps | | | | | | Clotho | | | | | |
| | | Text-to-audio | | | Audio-to-text | | | Text-to-audio | | | Audio-to-text | | |
| | | R@1↑ | R@5↑ | R@10↑ | R@1↑ | R@5↑ | R@10↑ | R@1↑ | R@5↑ | R@10↑ | R@1↑ | R@5↑ | R@10↑ |
|---|---|---|---|---|---|---|---|---|---|---|---|---|---|
| HTSAT-BERT (Baseline) | AC+Clotho | 39.2 | 74.9 | 86.5 | 49.5 | 81.9 | 91.5 | 15.6 | 38.4 | 52.0 | 21.0 | 43.8 | 55.7 |
| HTSAT-BERT-ZS | WavCaps | 28.6 | 61.1 | 75.8 | 40.2 | 69.4 | 80.3 | 16.5 | 38.8 | 50.9 | 20.0 | 43.3 | 56.6 |
| HTSAT-BERT-ZS | BEWO-1M | 23.6 | 56.3 | 70.8 | 28.2 | 57.9 | 71.6 | 12.0 | 31.0 | 42.3 | 12.6 | 28.4 | 40.0 |
| HTSAT-BERT-FT | WavCaps→AC+Clotho | 42.2 | 76.5 | 87.1 | 54.6 | 85.2 | 92.4 | 19.7 | 45.7 | 59.4 | 26.9 | 52.6 | 64.9 |
| HTSAT-BERT-FT | BEWO-1M→AC+Clotho | 41.6 | 77.3 | 87.7 | 53.8 | 83.9 | 92.9 | 18.5 | 43.0 | 56.3 | 21.0 | 43.7 | 57.4 |

Table G23: The evaluation results of 2-C audio-language retrieval on the test sets of BEWO-1M.

| Model | Training Data | Text-to-audio | | | Audio-to-text | | |
| | | R@1↑ | R@5↑ | R@10↑ | R@1↑ | R@5↑ | R@10↑ |
|---|---|---|---|---|---|---|---|
| HTSAT-BERT (Baseline) | WavCaps | 10.9 | 39.3 | 54.1 | 11.6 | 38.6 | 53 |
| HTSAT-BERT | BEWO-1M | 14.5 | 46.8 | 61.2 | 16.0 | 46.2 | 61.5 |

c) **Results and Analysis**: The 1-C retrieval results presented in Tab. G22 demonstrate that our dataset achieves a comparative performance to the popular models across all metrics, showcasing pre-training capabilities comparable to other large-scale datasets. In the dual-channel experiments, Tab. G23 shows that the 2-C retrieval model trained on BEWO-1M achieves superior retrieval performance compared to the one trained on WavCaps on the BEWO-1M test set. Notably, the BEWO-1M dataset and the retrieval model are not specifically designed for 2-C audio retrieval tasks; therefore, the experiments in this section provide sufficient proof for the extensive value and impact of our dataset.

### G.6 AUDIO CAPTIONING

Audio captioning aims to represent audio content using natural language descriptions. In this part, we follow the Automated Audio Captioning experiments in WavCaps.

a) **Models**: Audio captioning typically employs an encoder-decoder architecture. The audio encoder extracts acoustic features, which are then utilized by a language decoder to generate natural language captions. In our implementation, we adopted CNN14 from PANNs (Kong et al., 2020) as the audio encoder and a pre-trained BART (Lewis et al., 2019) model as the language decoder.

b) **Experimental Setup**: The setup of audio captioning is similar to our audio retrieval experiments. For single-channel data, we first train zero-shot models separately using WavCaps and BEWO-1M. Only 30% data of the BEWO-1M training set is used to train the model. Subsequently, we fine-tuned these pre-trained models on AudioCaps and Clotho datasets. The zero-shot models are trained for 15 epochs with a learning rate of $5 \times 10^{-5}$. Then, upon training on BEWO-1M, models are fine-tuned for 20 epochs with a learning rate of $5 \times 10^{-6}$. The final checkpoints are selected based on validation performance, and evaluation is performed on the test sets of AudioCaps and Clotho. We employ standard captioning metrics for evaluation, including BLEU (Papineni et al., 2002), ROUGE (Lin, 2004), METEOR (Banerjee & Lavie, 2005), CIDEr (Vedantam et al., 2015), SPICE (Anderson et al., 2016), and SPIDEr (Liu et al., 2017). For dual-channel evaluation, models are evaluated on the BEWO-1M test set. When evaluating models pre-trained on WavCaps, the audio input is converted to mono through channel averaging. The dual-channel audio captioning models are trained on 30% data of the BEWO-1M training set with minor adjustments on the baseline model.

c) **Results and Analysis**: As shown in Tab. G24, after fine-tuning, the 1-C model pre-trained on BEWO-1M demonstrates comparable performance to that pre-trained on WavCaps, with only marginal differences. This is particularly noteworthy given that WavCaps is specifically designed for 1-C audio captioning tasks.

However, in 2-C evaluations, Tab. G25 shows the model trained on BEWO-1M significantly outperforms that pre-trained on WavCaps.

Table G24: The evaluation results of 1-C audio captioning on the test sets of AudioCaps and Clotho. A higher score means better performance. "ZS" refers to zero-shot, "FT" refers to fine-tune, "A → B" means the model is pre-trained on dataset "A" and then fine-tuned on dataset "B".

| Model | Training Data | AudioCaps | | | | | | Clotho | | | | | |
|---|---|---|---|---|---|---|---|---|---|---|---|---|---|
| | | $BLEU_1\uparrow$ | $ROUGE_l\uparrow$ | METEOR↑ | CIDEr↑ | SPICE↑ | SPIDEr↑ | $BLEU_1\uparrow$ | $ROUGE_l\uparrow$ | METEOR↑ | CIDEr↑ | SPICE↑ | SPIDEr↑ |
| CNN-BART-ZS | WavCaps | 55.1 | 37.1 | 18.6 | 45.3 | 11.9 | 28.6 | 29.9 | 29.3 | 12.0 | 24.8 | 8.7 | 16.7 |
| CNN-BART-ZS | BEWO-1M | 15.8 | 18.9 | 7.5 | 22.6 | 6.5 | 14.5 | 10.3 | 13.5 | 5.8 | 7.6 | 3.7 | 7.6 |
| CNN-BART-FT | WavCaps→AC+Clotho | 69.3 | 49.9 | 24.7 | 75.6 | 17.9 | 46.8 | 60.1 | 40.0 | 18.5 | 48.8 | 13.3 | 31.0 |
| CNN-BART-FT | BEWO-1M→AC+Clotho | 63.4 | 44.9 | 20.7 | 57.2 | 15.6 | 36.4 | 54.9 | 36.4 | 16.7 | 38.5 | 11.4 | 25.0 |

Table G25: The evaluation results of 2-C audio captioning on the test sets of BEWO-1M.

| Model | Training Data | BEWO-1M test sets | | | | | |
|---|---|---|---|---|---|---|---|
| | | $BLEU_1\uparrow$ | $ROUGE_l\uparrow$ | METEOR↑ | CIDEr↑ | SPICE↑ | SPIDEr↑ |
| CNN-BART | WavCaps | 7.5 | 14.1 | 5.9 | 14.1 | 7.5 | 10.3 |
| CNN-BART | BEWO-1M | 31.9 | 35.3 | 16.3 | 45.1 | 22.5 | 33.8 |

## G.7 EVALUATION ON THE SCALE OF CLASSIFIER FREE GUIDANCE

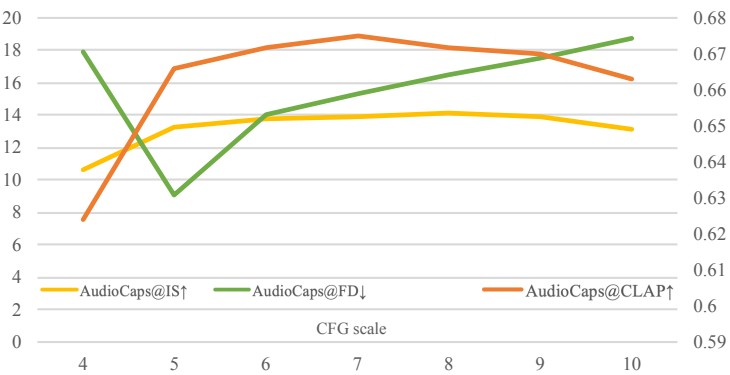

Figure G15: The impact of CFG scale. Overall the scale of 6 is recommended as the balance of the trade-off.

Due to the impact of different CFG scales on the generated audio, we conduct inference experiments using various CFG scales. As previously discussed, finding the most reasonable trade-off is crucial for achieving the best results. In Table. G26 and Fig. G15, our experiments demonstrate that a CFG scale of 6 provides relatively optimal outcomes. Thus, unless otherwise specified, we will set the CFG scale to 6 in all subsequent experiments.

## G.8 EVALUATION ON PERTAINING OF IMAGE ENCODER.

Since not all images have acoustic semantic captions, we conduct tests on the image dataset of BEWO-1M. The performance, pre-trained on our acoustic semantic captions, is shown in Tab. G.8 for reference. Since we only update the encoder for modality alignment, this performance might seem limited. However, we believe that the limited performance in visual captioning will not impede our model's ability to develop regional perception of images. Additionally, the boost in performance with regional perception further demonstrates the effectiveness of the strategy.

Table G26: The impact of CFG scale. Overall the setting of 6 is recommended as the balance of the trade-off.

| CFG scale $w$ | $w=4$ | $w=5$ | $w=6$ | $w=7$ | $w=8$ | $w=9$ | $w=10$ |
|---|---|---|---|---|---|---|---|
| AudioCaps@CLAP↑ | 0.624 | 0.666 | 0.672 | 0.675 | 0.672 | 0.670 | 0.663 |
| AudioCaps@IS↑ | 10.63 | 13.30 | 13.79 | 13.94 | 14.13 | 13.90 | 13.15 |
| AudioCaps@FD↓ | 17.89 | 9.07 | 14.01 | 15.31 | 16.53 | 17.50 | 18.74 |
| Mix-set@FSAD↓ | 0.147 | 0.153 | 0.160 | 0.167 | 0.168 | 0.191 | 0.243 |

Table G27: We use the metric of the caption generation to test the performance of the pre-trained image encoder and decoder.

| Image Captioning on BEWO-1M | CIDEr↑ | METEOR↑ | SPICE↑ |
|---|---|---|---|
| w/. regional perception | 68.3 | 21.7 | 17.3 |
| w/o. regional perception | 62.1 | 20.4 | 16.6 |

### G.9 SOURCE DIRECTION IMPACT ON THE AUDIO QUALITY

We maintain the overall caption and text length while only modifying the directional description in a few simple samples to assess the quality of the generated audio. Interestingly, we observe that a larger disparity between the ears (as the sound source moves further left or right) leads to a reduction in audio quality, as shown in Tab. G28. However, it is not due to any bias in the dataset regarding sound source locations because we control its proportion in simulated audio. We attribute this to the fact that the entropy of the latent code is limited, and the additional positional information competes with and displaces part of the original latent code, thereby slightly degrading the audio quality.

Table G28: A larger disparity between the ears (as the sound source moves further left or right) leads to a reduction in audio quality.

| Direction | ISc↑ | FD↓ | FAD↓ |
|---|---|---|---|
| Left | 10.72 | 18.70 | 5.08 |
| Front left | 11.01 | 18.34 | 5.15 |
| Front | 11.60 | 16.06 | 2.52 |
| Front right | 11.00 | 17.47 | 4.33 |
| Right | 10.86 | 18.80 | 4.75 |
| Moving | 10.48 | 18.51 | 4.89 |

### G.10 CAPTION LENGTH IMPACT ON THE AUDIO QUALITY

We conduct extensive generation on 5000 audios to generate simple dog barks, using GPT-4 to extend the length of captions with less meaningful words. We roughly analyze the relationship between caption length and the generated audio CLAP score. It is found that increases in caption length led to a significant decrease in the CLAP score in Table. G16. On one hand, longer captions pose challenges for the generation. On the other hand, the CLAP score does not accurately reflect the quality of longer captions. To determine the specific reasons, further analysis and statistics are required.

### G.11 POTENTIAL CLASS LEVEL BIAS

Inherent bias is a significant concern for professionals. We conduct further observations to gain deeper insights. Before we begin the analysis, let's clarify this important question. For instance, when a piano is recorded, the output from the LLM is typically static and indoors. In contrast, when it comes to living beings like humans or dogs, the LLM often assumes they are moving and outdoors. Therefore, this is a potential inherent bias in the data and the LLM itself.

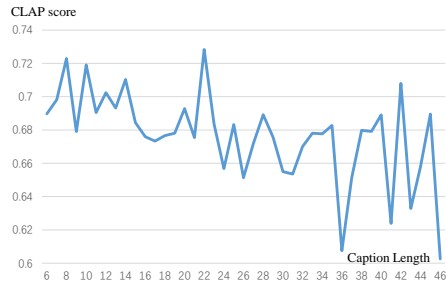

Figure G16: The chart of caption length causing a decrease and fluctuating on CLAP score.

Therefore, We evaluated five common classes individually. Only minor differences between them are observed, likely due to complex reasons like data scale and individual differences. Generally, everything appears normal except for the generation of airplane echo sounds. It seems that LLM in data collection consistently assumes that airplanes cannot exist in a small Reverberating room, thereby introducing a specific type of bias.

It is interesting and somehow controversial. Is this bias good or bad? We are trying to use "rationality" to avoid unrealistic scenarios like "*an Airplane is passing by in the small reverberating room*". In this case, this scenario is unrealistic and unnecessary at all. So, it could be a coin of two sides. But as a pioneer work, we still prove that most of the common objects do not suffer from such bias. The rest of this potential bias is more than welcome to explore in the future.

Table G29: To observe the potential class-wise bias, we select 5 common classes and evaluate each of them separately. Reverberation and distance means the special factor described in prompts and ✔ means the ability to control.

| Target Classes | Stationary (FSAD) | Moving (FSAD) | Reverberation | Distance |
|---|---|---|---|---|
| Dog | 0.141 | 0.137 | ✔ | ✔ |
| Man speaking | 0.170 | 0.177 | ✔ | ✔ |
| Airplane | 0.179 | 0.164 | ✗ | ✔ |
| Piano | 0.243 | 0.215 | ✔ | ✔ |
| Violin | 0.267 | 0.251 | ✔ | ✔ |

### G.12 VISUALIZATION OF COARSE AND FINE GUIDANCE

In addition to using captions, we also use attributes to guide audio generation. Fig. G17 visualizes the guidance. By defining speed, azimuth, moving time, etc., we can fine-tune and infer guidance.

## H PROMPTS USED IN DATA CONSTRUCTION AND INFERENCE

We design several prompts for GPT-4 and GPT-4o for captions transformation and attribute inference.

- Tab.H30: Dataset construction of audio-caption pair. We provide captions and pre-selected attributes (start/end positions and movements) to GPT-4. Then we require other attributes (scene size and speed) and the transformed caption.

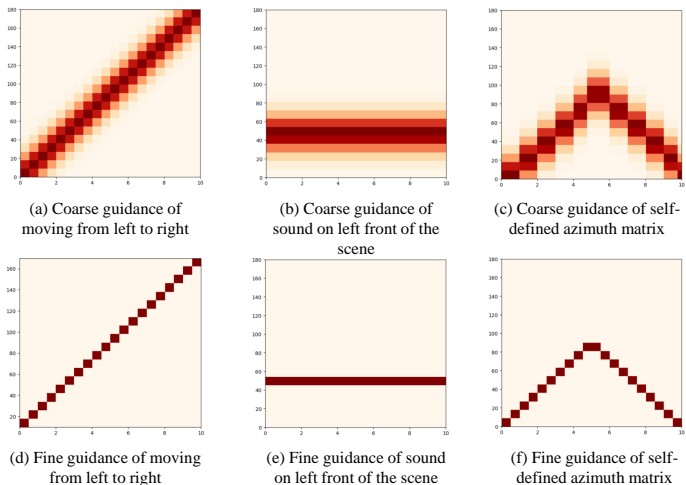

(a) Coarse guidance of moving from left to right

(b) Coarse guidance of sound on left front of the scene

(c) Coarse guidance of self-defined azimuth matrix

(d) Fine guidance of moving from left to right

(e) Fine guidance of sound on left front of the scene

(f) Fine guidance of self-defined azimuth matrix

Figure G17: The visualization of azimuth state matrix before normalization. Time for $x$-axis and angle for $y$-axis. Through this guidance, we can define speed, azimuth, moving time, etc.

- Tab.H31: Dataset construction of image-caption pair. We provide a caption, an image, and a position list of detected objects. Then we require sound objects with attributes (scene size, start/end position, and speed) and the transformed caption.

- Tab.H32: Inference with caption input. We provide a caption to GPT-4. Then we require sound objects with attributes (scene size, start/end positions, and speed).

- Tab.H33: Inference and interaction with image input. We provide an image and a position list of detected objects. Then we require sound objects with attributes (scene size, start/end position, and speed) and the transformed caption.

**Data construction of audio-caption pair**

I will provide you with captions and some attribute lists to describe an audio clip.

The caption and elements in the lists are in the same order. You have to return a new caption and other attributes - scene size and speed based on my input. Please follow the procedures step-by-step strictly!

1) Assess the size of the scene in which the audio occurs from: - Scene size choices: '1:outdoors', '2:large', '3:moderate', '4:small'.

2) Identify objects' sounds description and add their init directions from input - Basic description: 'laughing', 'Speaking', 'Meowing', 'Blowing', 'Pouring' ...

3) Read the movement of objects from input. If not moving, skip step 4.

4) If the objects are moving, also read their end directions from input, and choose speeds for the moving object from: - Speed choices: '1:slow', '2:moderate', '3:fast', '4:instantly'.

5) If the speed is '4:instantly', you need to use a special method to rewrite the caption. It's not an object moving fast, you have to think of there are two sound sources sounding sequentially from different positions. For example, if input is "a dog barks"; movement:["moving"]; init direction:["left"]; end direction:["right"]. And you think its speed is "instantly". Its caption should be like "a dog barks at left, then another dog barks at right".

6) Ensure that the moving speed and scene size you choose correspond realistically with objects in the real world.

7) Brief me only 1 overall sound description of the possible audio with appropriate direction words. Try to use less than 30 words for the caption. And ignore the other things that cannot produce sound.

8) All objects should be described with direction and movement. If it's still, you can drop the movement words. If an object has no sound description, you have to infer a reasonable sound for it.

9) Provide your response in JSON format beginning with "{" like the examples below.

Example: with input as: caption: ["bus idles", "woman"]; init direction: ["left", "right"]; movement:["no moving", "moving"]; end direction:["left","directly front"]. You should respond: {"size": 1,"objects": {"Bus": {},"Woman": {"speed":1}, ...}, "one_sentence_brief_only_audio_caption": "The bus engine idles on the left and a woman walks from right to front slowly."}

---

**Input**

Caption: ["A dog is barking","guitar"]; init direction: ["front", "right"]; movement:["no moving", "moving"]; end direction:["front","front left"].

---

**Output**

{ "size": 2, "objects": { "Dog": { }, "Guitar": { "speed": 2 } }, "one_sentence_brief_only_audio_caption": "A dog barks in front while a guitar strums from right to front left moderately." }

Table H30: The prompt and an example in data construction based on audio-caption pair.

**Data construction of image-caption pair**

I will provide you with an image with its caption, and a list of positions of objects.
The position list records the locations of some objects,
the top left corner is (0, 0) and the bottom right is (1,1).
You have to return attributes and a new caption based on my input.
Please follow the procedures step-by-step strictly!
1) Determine if the entire scene is likely to produce sound.
Based on the picture and caption, identify the objects that may produce sound. If one object can make a sound, you can think it can sound. If impossible, skip other steps. - Sounding choices: '0: impossible', '1: possible'
2) Assess the size of the scene in which the audio occurs from: - Scene size choices: '1:outdoors', '2:large', '3:moderate', '4:small'.
3) Identify the objects whose positions are in the position list, and think in the original order of the list. Discard objects that cannot produce sound and those not in the position list. Then add the position with basic descriptions for each object based on the position list. The descriptions of direction and distance are based on the absolute position, not the positional relationship between objects. For example, (0.3, y) should be described as front left. You should consider both direction and distance: - direction choices: '1:left', '2:front left', '3:directly front', '4:front right', '5:right'. - Distance choices: '1:far', '2:moderate', '3:near'. - Basic description: 'laughing', 'Speaking', 'Meowing', 'Blowing', 'Pouring' ...
4) Identify if the object is moving or not. If not, skip step 5. - Moving choices: '0: No moving', '1: Moving'
5) If some objects move, choose an end position that is different from the init direction, and also choose a speed for the object from: - Speed choices: '1:slow', '2:moderate', '3:fast'.
6) For both init direction and end direction, note that you can return a decimal number to make result precise based on the input position. You can simply think the front is 3.0, left is 1.0 and right is 5.0. Thus, with input position (0.3, y), its direction is 2.2.
7) Ensure that the positions, movements, and scene size you choose correspond realistically with objects in the real world.
8) Brief me only 1 overall sound description of the possible audio with appropriate direction and movement words. Try to use less than 30 words for the caption. And ignore the other things that cannot produce sound. All the objects should consider the position list, raw caption, and image input.
9) All objects should be described with direction and movement. If it's still, you can drop the movement words. You are only required to describe distance if it's at an extreme distance.
10) Provide your response in JSON format beginning with "{" like the examples below.
Example: {"sound": 1,"size": 1,"objects": {"Bus": {"init_direction": 1, "init_dis": 1, "moving":0,}, "Woman": {"init_direction": 5, "init_dis":1, "moving":1, "end_direction": 3, "end_dis":2, "speed":1}}, "one_sentence_brief_only_audio_caption": "The bus engine idles on the left and a woman walks from right to front slowly.",}

---

**Input**

Caption: "the large bird is flying threw the air."; Position: [(0.35, 0.25)]

---

**Output**

{ "sound": 1, "size": 1, "objects": { "Bird": { "init_direction": 2.2, "init_dis": 2, "moving": 1, "end_direction": 3.5, "end_dis": 1, "speed": 2 } }, "one_sentence_brief_only_audio_caption": "The bird flaps its wings from front left to directly front at a moderate speed." }

Table H31: The prompt and an example in data construction based on image-caption pair.

**Inference with caption input**

I will provide you with a caption. You have to return attributes based on my input.

Please follow the procedures step-by-step strictly!

1) Determine if the entire scene is likely to produce sound. Based on the input, identify the objects that may produce sound. If one object can make a sound, you can think it can sound. If impossible, skip other steps. - Sounding choices: '0: impossible', '1: possible'

2) Assess the size of the scene in which the audio occurs from: - Scene size choices: '1:outdoors', '2:large', '3:moderate', '4:small'.

3) Identify objects' sound descriptions. Then determine the position and basic descriptions for each object based on the input. You should consider both direction and distance: - direction choices: '1:left', '2:front left', '3:directly front', '4:front right', '5:right'. - Distance choices: '1:far', '2:moderate', '3:near'.

4) Identify if the object is moving or not. If not, skip step 5. - Moving choices: '0: No moving', '1: Moving'

5) If some objects move, choose an end position that is different from the init direction, and also choose a speed for the object from: - Speed choices: '1:slow', '2:moderate', '3:fast'.

6) For both init direction and end direction, note that you can return a decimal number when input involves a precise description of the angle. e.g. if the input is "a dog barks at front left 70 degree", you can return "init_direction": 2.7, or "init_direction": 2.8, and you can infer what makes sense.

7) Ensure that the positions, movements, and scene size you choose correspond realistically with objects in the real world and match the input.

8) Provide your response in JSON format beginning with "{" like the examples below.

Example: with input as: "The bus engine idles on the left and a woman walks from right to front slowly.". You should respond: {"sound": 1, "size": 1, "objects": {"Bus": {"init_direction": 1, "init_dis": 1, "moving":0,}, "Woman": {"init_direction": 5, "init_dis":1, "moving":1, "end_direction": 3, "end_dis":2, "speed":1}}

---

**Input**

A man speaks in front while a dog barks from front right to left.

---

**Output**

{ "sound": 1, "size": 3, "objects": { "Man": { "init_direction": 3, "init_dis": 3, "moving": 0 }, "Dog": { "init_direction": 4, "init_dis": 3, "moving": 1, "end_direction": 1, "end_dis": 2, "speed": 2 } } }

Table H32: The prompt and an example in text inference.

**Inference and interaction with image input**

I will provide you with an image and a list of objects' positions.
The position list records the locations of some objects,
the top left corner is (0, 0) and the bottom right is (1,1).
You have to return attributes and a caption based on my input.
Please follow the procedures strictly step-by-step!

1) Determine if the entire scene is likely to produce sound.
Based on the picture, identify the objects that may produce sound. If one object can make a sound, you can think it can sound. If impossible, skip other steps. - Sounding choices: '0: impossible', '1: possible'
2) Assess the size of the scene in which the audio occurs from: - Scene size choices: '1:outdoors', '2:large', '3:moderate', '4:small'.
3) Identify the objects whose positions are in the position list, and think in the original order of the list. Discard objects that cannot produce sound and those not in the position list. Then add the position with basic descriptions for each object based on the position list. The descriptions of direction and distance are based on the absolute position, not the positional relationship between objects. For example, (0.3, y) should be described as front left. You should consider both direction and distance: - direction choices: '1:left', '2:front left', '3:directly front', '4:front right', '5:right'. - Distance choices: '1:far', '2:moderate', '3:near'. - Basic description: 'laughing', 'Speaking', 'Meowing', 'Blowing', 'Pouring' ...
4) Identify if the object is moving or not. If not, skip step 5. - Moving choices: '0: No moving', '1: Moving'
5) If some objects move, choose an end position that is different from the init direction and distance, and also choose a speed for the object from: - Speed choices: '1:slow', '2:moderate', '3:fast'.
6) For both init direction and end direction, note that you can return a decimal number to make result precise based on the input position. You can simply think the front is 3.0, left is 1.0 and right is 5.0. Thus, with input position (0.3, y), its direction is 2.2.
7) Ensure that the positions, movements, and scene size you choose correspond realistically with objects in the real world.
8) Brief me only 1 overall sound description of the possible audio with appropriate direction and movement words. Try to use less than 30 words for the caption. And ignore the other things that cannot produce sound. All the objects should consider the position list and image input.
9) All objects should be described with direction and movement. If it's still, you can drop the movement words. You are only required to describe distance if it's at an extreme distance.
10) Provide your response in JSON format beginning with "{" like the examples below.
Example: {"sound": 1,"size": 1,"objects": {"Bus": {"init_direction": 1, "init_dis": 1, "moving":0,}, "Woman": {"init_direction": 5, "init_dis":1, "moving":1, "end_direction": 3, "end_dis":2, "speed":1}}, "one_sentence_brief_only_audio_caption": "The bus engine idles on the left and a woman walks from right to front slowly."}

---

**Input**
Position:[(0.6, 0.3),(0.4, 0.7)]

---

**Output**
{ "sound": 1, "size": 3, "objects": { "Bird1": { "init_direction": 4.2, "init_dis": 2, "moving": 0 }, "Bird2": { "init_direction": 2.8, "init_dis": 2, "moving": 0 } }, "one_sentence_brief_only_audio_caption": "Birds chirp softly from the front left and front right." }

Table H33: The prompt and an example in image inference.

# I    FREQUENT ASKED QUESTIONS

## I.1    CLAP SCORE ON GT

Based on our experiments, the CLAP score for the ground truth audio in the AudioCaps test set is 0.67, which is the same as Huang et al. (2023a). The process of converting audio to description involves some information loss, which means that certain sounds in the test set may lack appropriate descriptions, resulting in a ground truth CLAP score of less than 1. However, advanced models like TANGO 2, Audio-Box, and SpatialSonic (ours) are currently able to achieve CLAP scores that approach or even exceed the ground truth. This is a common result of these models accurately following text context. And we are willing and excited to open our checkpoints and audio samples for public verification.

## I.2    DIFFERENCE BETWEEN STEREO AND SPATIAL AUDIO

Stereo audio primarily uses two channels to create a sense of width by directing different sounds to each ear. However, it can be extended to include more channels in certain setups, enhancing the audio experience with additional depth and dimension. This allows listeners to perceive sounds from various directions, including above and below, often using formats like 5.1 or 7.1 surround sound (Herre et al., 2015a;b) for a more immersive experience. Both stereo and spatial audio aim to create an immersive listening experience by simulating the natural way we perceive sound in our environment. In this work, we focus solely on azimuth as a metric for stereo and spatial audio. In this context, we equate stereo audio and spatial audio.

## I.3    ELEVATION AND FRONT-BACK CONFUSION

In audio localization research, sound source localization is divided into two dimensions: azimuth and elevation. Researchers use more microphones to balance these dimensions (Jekateryńczuk & Piotrowski, 2024). In our simulation, we use a 2-channel omnidirectional microphone array with consistent sensitivity in all directions, which is designed for widely-used dual microphone devices. This microphone array in most of the simulators cannot simultaneously localize azimuth and elevation (Jekateryńczuk & Piotrowski, 2024) and experiences front-back confusion (van der Heijden & Mehrkanoon, 2022; van der Heijden et al., 2019; Orr et al., 2023). Therefore, when constructing the BEWO-1M dataset, we do not consider elevation or front-back, assuming the sound source to be within the front 180° range in azimuth. Adding more microphones, using microphones with artificial human-shaped ears (Yang & Zheng, 2022) and the Head-Related Transfer Function (HRTF) algorithm can reduce these confusions (Orr et al., 2023) but bring challenges to large-scale simulations. Our work proposes a data synthesis and model pipeline, which can be adapted for more types of simulations and generations in the future, potentially requiring more detailed captions.

## I.4    DISTANCE CONTROLLING AND EMBEDDING

The previous See2Sound used the Depth-Anything model for depth perception. However, our model does not specifically encode the distance attribute in azimuth state matrices. This is because we require GPT-4 or GPT-4o to describe it in the text when distance is crucial. The distance information is directly presented in the modality embedding without requiring additional encoding in the azimuth matrix.

# J    REVISION LOG

Specifically, we have included the revision log here with the date format of `dd/mm/yyyy`. We welcome any suggestions that may contribute to the improvement of this paper.

### J.1 BEWO-1.1: 11/17/2024

1. Better Presentation

   (a) Revision of the summary of the dataset construction pipeline in Sec. 3 and the model framework in Sec. 4.

   (b) Revision of image-related part in Sec. 4.2 and Sec. 4.4 for better integration.

   (c) Replace MAE to MSE. Sorry about the mistake.

2. Additional Experiments

   (a) Inclusion of supplementary experiments utilizing the less usual dataset **Clotho**, detailed in Section G.1.

   (b) Conducting further experiments on **text-to-audio retrieval** and **audio captioning**, presented in Sections G.5 and G.6 respectively. These are included even though the primary focus of the paper remains on generation tasks.

3. Insights and Analysis

   (a) Integration of additional analysis regarding the **cfg scale**, elaborated in Section G.7.

   (b) Provision of further insights into the potential **class-level bias**, which is beneficial for the community, discussed in Section G.11.

   (c) Extension of analysis concerning the **FSAD**, documented in Section F.3.

4. Formatting Adjustments

   (a) Minor modifications in Section 2 to ensure the elimination of the widow words.

### J.2 POTENTIAL REQUIREMENTS

This section is used to provide the community with maximum effort on spatial audio generation.

Please feel free to contact us at any time if you have any requests from the list below. We will make the requested version of the data available as open-source.

- BEWO-1M with $44.1K$ Hz.
- BEWO-1M with $5.1$-channel microphone array.

