# OpenReview forum: "Both Ears Wide Open: Towards Language-Driven Spatial Audio Generation"
_ICLR.cc/2025/Conference — ICLR 2025 Spotlight_

### Official Review · Reviewer_VEpT · 2024-11-03

**Soundness:** 3
**Presentation:** 4
**Contribution:** 4
**Rating:** 8
**Confidence:** 4

**Summary:**

The paper titled "Both Ears Wide Open: Towards Language-Driven Spatial Audio Generation" presents a novel approach in the field of audio generation, focusing on the creation of stereo audio that adheres to spatial contexts described by text or images. The authors address the challenges of generating controllable spatial audio by introducing a large-scale dataset, BEWO-1M, and a model named SpatialSonic. This model leverages spatial-aware encoders and azimuth state matrices to generate immersive and directional audio in line with textual and visual prompts. The paper also discusses the subjective and objective evaluations conducted to validate the effectiveness of the proposed method against existing approaches. Overall, I think the task definition of this paper is good, the data collection process and model design are excellent, the paper writing and overall appearance are good, and it deserves a higher score.

**Strengths:**

1. The paper introduces a pioneering dataset, BEWO-1M, which contains a rich collection of audio samples with spatial descriptions, significantly advancing the field of spatial audio generation.
2. The SpatialSonic model employs a sophisticated design that integrates multimodal inputs (text and images) to produce spatial audio, demonstrating a high degree of control over audio generation.
Writing Ideas: The paper is well-structured, with a clear problem statement and a logical progression from dataset creation to model development and evaluation, making it easy to follow the authors' line of reasoning.
3. The authors conduct both subjective and objective evaluations, providing a comprehensive assessment of the model's performance. In particular, the stereo effect in the demo provided in this article is amazing, which will have a wide range of applications in real life.

**Weaknesses:**

1. The authors "apply sound activity detection on each audio and randomly crop to 10s segments.", This method seems a bit random, will it affect the quality of the original audio data?
2. According to the data collection pipeline in Figure 2, it seems easy to collect audio-text data paired with image data, but why is there so little image data in Table 1?
3. Taking video as a data source and incorporating it into the process of dataset collection and model design will make this task more promising.

**Questions:**

See [Weaknesses].

---

> ### Author Response · Authors · 2024-11-21
> **To #VEpT**
>
> To #VEpT: Thanks for your recognition of the novel method, well-organized structure, and immersive demo. We feel highly encouraged and make the following efforts to address your concerns.
>
> 1. Random in data collection pipeline.
>     1. Explaination: Such cropping is a **common practice** in T2A like Make-An-Audio[1]. Additionally, we follow the WavCaps to filter the cropped audio segments based on their CLAP scores. This ensures that cropped audio segments contain sufficient information to match their captions.
>     2. Observation experiment: We design a subjective evaluation comparing the cropped audio segments with their corresponding original audio. We randomly select 1,000 audio pairs from the dataset source and divide them into 5 groups. Testers are asked to rate MOS (Mean Opinion Score) based on overall audio quality. The results demonstrate that our cropping method does not affect the quality of the original audio data.
>
>         | Fold   | 1    | 2    | 3    | 4    | 5    | AVG  |
>         |--------|------|------|------|------|------|------|
>         | Origin | 4.96 | 4.98 | 4.92 | 4.98 | 4.95 | 4.96 |
>         | Crop   | 4.95 | 4.95 | 4.94 | 4.97 | 4.96 | 4.95 |
>
> 2. The amount of image data (revised in Sec.4.2).
>     1. Data: Coco-2017, as the most common detection and captioning dataset, is used as the image source of BEWO-1M. Some of the classes in Coco-2017 can not make a sound at all, like table, spoon, fork… So we have to apply a **selection strategy** for clean data. Moreover, the retrieval in the data collection pipeline needs a rather strict threshold to ensure accuracy.  After all the preprocessing, we still have far exceeded the number of images in the commonly used dataset Imagehear [2] (100 images). Considering the problem of limited image data, we use up to 10 audio clips paired with each image to prevent the generation collapse in Line 963.
>     2. Model: Overall, we tend to use the **language-driven** method to build up the generation model, entitled “Language-Driven”. For better clarity, we revise the statement in Sec.4.2 to emphasize that the image encoder is to align the image to make full use of the important knowledge from audio-text pairs. Such alignment, within the Coco-2017, does **not require** the same amount of data as text. As shown in Tab.4 and Tab.8, our model achieves comparative results on Coco-2017.
> 3. Video2Audio (V2A).
>     * The suggestion you mentioned was indeed considered a few months ago. Temporal modeling stands out as a critical aspect of Video2Audio generation. For instance, accurately aligning a video of a bouncing basketball with its "pounding" is essential. While it is feasible with a sophisticated, well-designed **temporal perception module**, we struggled to condense the current manuscript to fit within the 10-page limit, **even without the video**. Besides, we are working on Video2Audio and trying to solve this temporal alignment problem in an more elegant way. Therefore, such a different task will not suppress our contribution to the pioneering datasets, models, and evaluation methods.
>
> *[1] Huang, R., Huang, J., Yang, D., Ren, Y., Liu, L., Li, M., ... & Zhao, Z. (2023, July). Make-an-audio: Text-to-audio generation with prompt-enhanced diffusion models. In ICML (pp. 13916-13932). PMLR.*
>
> *[2] Sheffer, R., & Adi, Y. (2023, June). I hear your true colors: Image-guided audio generation. In ICASSP (pp. 1-5). IEEE.*

---

> ### Author Response · Authors · 2024-11-26
> **Please let us know whether all issues are addressed**
>
> Thanks for the comments. We have provided more explanations and answers to your questions. Since the deadline for discussion is near the end, please let us know whether we have answered all the questions.
>
> Best regards,
>
> Authors

---

### Official Review · Reviewer_pDcX · 2024-11-04

**Soundness:** 3
**Presentation:** 2
**Contribution:** 3
**Rating:** 6
**Confidence:** 3

**Summary:**

This paper propose a new dataset and method for stereo audio generation. Proposed method can generate spatial audio from the instruction format of text, image or interactives. The model are proven effective from the sota comparison tabels.

**Strengths:**

1. The sterro audio generation is less studied than general mono channel audio generation but is meaningful for immersive experiences, and is a good research topic.
2. Proposed dataset BEWO-1M is the first large-scale spatial audio dataset which is labeled through automatic pipelines, the construction pipeline sounds reasonable and I believe the dataset could be helpful for future research of this field.
3. Proposed method are proven effective on both mono-channel benchmarks and proposed dual-channel benchmark,

**Weaknesses:**

1. The descriptions of dataset construction pipeline and model framework are not easy to follow enough.

**Questions:**

1. I would like to see if BEWO-1M could be a better dataset for audio-language pretraining than current datasets, such as WavCaps, on tasks like text-to-audio retrieval or audio captioning? The inferior conclusion  is acceptable due to that this dataset is not designed targeted for those tasks.
2. In the stereo audio generation benchmark evaluation, arthors also compared their model with AudioLDM2, Make-An-Audio2, etc, the same as mono-channel benchmarks. Are there not any comparable method that also specialize in stereo audio generation?

---

> ### Author Response · Authors · 2024-11-21
> **To #pDcX (Part 1)**
>
> To #pDcX: Thank you for your positive statements about our dataset, model, and benchmark. We have dedicated a lot of effort over the past few days to provide the following insights for the community as requested.
>
> 1. The descriptions of the dataset pipeline and model framework (updated in Sec.3 and Sec.4).
>     * Our paper covers a wide range of contents, including the collection of data, as well as the training and evaluation of models. To maintain balance, we have to move some content to the Appendix; therefore, we have included a detailed 20-page appendix, especially from Sec.B-F. Furthermore, to address your concerns, we **revise** the beginnings of Sec.3 and Sec.4 to clarify the structure.
> 2. Result of text-to-audio retrieval or audio captioning (updated in Appendix.Sec.G5 and Appendix.Sec.G6).
>     * Experiment: Thanks for the tips to increase the influence of BEWO-1M. We supplement our paper with additional experiments on audio-language retrieval and captioning tasks, which now can be found in Appendix G.5 and G.6. We follow the experiments from WavCaps [1]. The retrieval model consists of an audio encoder (HTSAT [2]) and a language encoder (BERT [3]); meanwhile, the caption model consists of an audio encoder (CNN from PANNs[4]) and a language decoder (BART [5]). In the tables, “ZS” refers to zero-shot, and “FT” refers to fine-tuning. “A → B” in the Training Data column means the model is pre-trained on the dataset “A” and then fine-tuned on dataset “B”.
>     * Analysis: The 1-C retrieval and captioning results show that BEWO-1M achieves **competitive** pre-training performance compared to other large-scale datasets (e.g., WavCaps), even though our dataset is not specifically designed for these tasks. When evaluated on our stereo BEWO-1M test set, the models trained on BEWO-1M demonstrate superior performance compared to those trained on WavCaps.
>
>     Audio retrieval on 1-channel data in the table below.
>
>     | Model                 | Training Data       | AudioCaps(AC) |   AC   |   AC   |   AC   | Clotho  |  Clotho   |  Clotho  |   Clotho  |
>     | --------------------- | ------------------- | --------- | ------- | ------- | ------- | ------- | ------- | ------- | ------- |
>     |                       |                     | T2A R@1   | T2A R@5 | A2T R@1 | A2T R@5 | T2A R@1 | T2A R@5 | A2T R@1 | A2T R@5 |
>     | HTSAT-BERT (Baseline) | AC+Clotho           | 39.2      | 74.9    | 49.5    | 81.9    | 15.6    | 38.4    | 21.0    | 43.8    |
>     | HTSAT-BERT-ZS         | WavCaps             | 28.6      | 61.1    | 40.2    | 69.4    | 16.5    | 38.8    | 20.0    | 43.3    |
>     | HTSAT-BERT-ZS         | BEWO-1M             | 23.6      | 56.3    | 28.2    | 57.9    | 12.0    | 31.0    | 12.6    | 28.4    |
>     | HTSAT-BERT-FT         | WavCaps → AC+Clotho | 42.2      | 76.5    | 54.6    | 85.2    | 19.7    | 45.7    | 26.9    | 52.6    |
>     | HTSAT-BERT-FT         | BEWO-1M → AC+Clotho | 41.6      | 77.3    | 53.8    | 83.9    | 18.5    | 43.0    | 21.0    | 43.7    |
>
>
>     Audio retrieval on 2-channel data of BEWO-1M test set in the table below.
>
>     | Model                 | Training Data | T2A R@1  | T2A R@5  | A2T R@1  | A2T R@5  |
>     | --------------------- | ------------- | -------- | -------- | -------- | -------- |
>     | HTSAT-BERT (Baseline) | WavCaps       | 10.9     | 39.3     | 11.6     | 38.6     |
>     | HTSAT-BERT            | BEWO-1M       | **14.5** | **46.8** | **16.0** | **46.2** |
>
>
>     Audio captioning on 1-channel data in the table below.
>
>     | Model                  | Training Data       | AC |  AC  |  AC | AC | AC |  AC  | Clotho |  Clotho |  Clotho |  Clotho| Clotho |    Clotho  |
>     | ---------------------- | ------------------- | --------- | ----- | ------ | ----- | ----- | ------ | ------ | ----- | ------ | ----- | ----- | ------ |
>     |                        |                     | BLEU1      | ROUGE-l | METEOR | CIDEr | SPICE | SPIDEr | BLEU1   | ROUGE-l | METEOR | CIDEr | SPICE | SPIDEr |
>     | CNN-BART-ZS  | WavCaps             | 55.1      | 37.1  | 18.6     | 45.3  | 11.9   | 28.6   | 29.9   | 29.3  | 12.0   | 24.8  | 8.7  | 16.7   |
>     | CNN-BART-ZS            | BEWO-1M             | 15.8      | 18.9  | 7.5    | 22.6  | 6.5   | 14.5   | 10.3   | 13.5  | 5.8    | 7.6   | 3.7   | 7.6    |
>     | CNN-BART-FT  | WavCaps → AC+Clotho | 69.3      | 49.9  | 24.7   | 75.6  | 17.9  | 46.8   | 60.1   | 40.0  | 18.5   | 48.8  | 13.3  | 31.0   |
>     | CNN-BART-FT            | BEWO-1M → AC+Clotho | 63.4      | 44.9  | 20.7   | 57.2  | 15.6  | 36.4   | 54.9   | 36.4  | 16.7   | 38.5  | 11.4  | 25.0  |

---

> ### Author Response · Authors · 2024-11-21
> **To #pDcX (Part 2)**
>
> (follow the upper block of Q2) Audio captioning on 2-channel data of BEWO-1M test set in table below.
>
>    | Model             | Training Data | BLEU1    | ROUGE-l | METEOR | CIDEr | SPICE | SPIDEr  |
>    |-------------------|---------------|---------|-------|--------|-------|-------|---------|
>    | CNN-BART-Baseline | WavCaps       | 7.5     | 14.1  | 5.9    | 14.1  | 7.5   | 10.3    |
>    | CNN-BART          | BEWO-1M       | **31.9**    | **35.3**  | **16.3**   | **45.1**  | **22.5**  | **33.8**    |
>
>
> 3. Comparison of other stereo audio generation models.
>     * We are the **first work** in the field of controllable spatial audio generation with the recognition of other reviewers. Nonetheless, we diligently compare our model with **all** relevant spatial audio models to the best of our knowledge in Sec.D, including but not limited to Stable-audio-open, See2sound, and Sepstereo.
>
> *[1] Mei, X., Meng, C., Liu, H., Kong, Q., Ko, T., Zhao, C., ... & Wang, W. (2024). Wavcaps: A ChatGPT-assisted weakly-labeled audio captioning dataset for audio-language multimodal research. IEEE/ACM Transactions on Audio, Speech, and Language Processing.*
>
> *[2] Chen, K., Du, X., Zhu, B., Ma, Z., Berg-Kirkpatrick, T., & Dubnov, S. (2022, May). Hts-at: A hierarchical token-semantic audio transformer for sound classification and detection. In ICASSP 2022-2022 IEEE International Conference on Acoustics, Speech and Signal Processing (ICASSP) (pp. 646-650). IEEE.*
>
> *[3] Devlin, J., Chang, M., Lee, K., & Toutanova, K. (2019). BERT: Pre-training of Deep Bidirectional Transformers for Language Understanding. North American Chapter of the Association for Computational Linguistics.*
>
> *[4] Kong, Q., Cao, Y., Iqbal, T., Wang, Y., Wang, W., & Plumbley, M. D. (2020). Panns: Large-scale pretrained audio neural networks for audio pattern recognition. IEEE/ACM Transactions on Audio, Speech, and Language Processing, 28, 2880-2894.*
>
> *[5] Lewis, M. (2019). Bart: Denoising sequence-to-sequence pre-training for natural language generation, translation, and comprehension. arXiv preprint arXiv:1910.13461.*

---

> ### Author Response · Authors · 2024-11-26
> **Please let us know whether all issues are addressed**
>
> Thanks for the comments. We have provided more explanations and answers to your questions. Since the deadline for discussion is near the end, please let us know whether we have answered all the questions.
>
> Best regards,
>
> Authors

---

### Official Review · Reviewer_Ph1d · 2024-11-04

**Soundness:** 3
**Presentation:** 3
**Contribution:** 2
**Rating:** 8
**Confidence:** 3

**Summary:**

This work explores spatial audio generation using latent diffusion. The authors propose to do this using a new 1M stereo audio dataset with paired captions which are LLM generated longform descriptions based on detected sound events where the LLM (GPT4*) includes explicit inferences of distance indicators like 'far/near' and room size; these are then used with a spatial audio simulator. For spatial audio generation, the authors then propose a conditional audio generation setup using text, image and positional (azimuth) conditions. The diffusion transformer model is essentially a fine-tuning on top of StabilityAI's audio work with defined conditions (768d) and using classifier-free guidance. Several ablations tracking both subjective (N=15) and classical audio generation metrics is then performed to assess quality; where the authors also introduce a twist on Frechet metrics to account for stereo-out.

**Strengths:**

* The problem itself is novel one as I don't think I've seen other works tackle the spatial audio generation problem in the same modern setup, e.g. via LDM.
* I think the dataset portion was the most interesting and seemingly rigorous part of the work and the spatial audio generation was the least. The dataset collection pipeline in general seemed like a sensible approach, short of collecting it manually, whereas the spatial audio generation portion felt like a fairly straightforward application. Good to see the authors' intent to make the data available to the community.
* I'm neutral on the claimed contribution of the metrics front since I didn't fully understand the benefits from the presentation (see question).

**Weaknesses:**

* I think the biggest challenge I had was that I was not qualitatively convinced of the quality from the demos. There were maybe only 2-3 samples in total where it felt like it was doing what was expected spatially, eg: "A duck is quacking on the left." or "A dog is barking on the right." don't actually sound like they're coming from their respective directions. That said, the framing of the paper's subjective results is that it is better _relative_ to competing methods, which is possible, but it feels like the differentiated delta is not perceptibly large. My guess is that this actually largely has to do with the way that the simulation works w/ room acoustics.
* I think there are likely still issue with using GPT to infer audio attributes, but at the same time the authors attempted to address this using manual validation.
* The whole image conditioning task honestly felt like a distraction to the rest of the paper. I would've just moved it into the appendix entirely and focused much more on the quality of spatial audio generation (text- / azimuth-conditioned) and additional ablation studies.

**Questions:**

* How critical was classifier-free guidance? I wonder to what degree of spatial audio quality was affected by this.
* Can the authors expand on the FSAD metric definition? How it's computed and why it is better suited to measure in the stereo-out setting? The section on this was so brief that I didn't completely understand it. It would help in the expansion to explicitly articulate why/how it's better than other metrics, and e.g. how meaningful percent improvements are.
* It would help to break performance differences by certain sound categories, e.g. it's reasonable to expect an "engine sound" or "dog bark" moving left-to-right, however, it perhaps might be less likely to expect that with "piano playing". This relates also to the audio simulation setup (moving audio sources vs moving microphones) and priors that come from the LLM approach (e.g., will "piano playing" performance always be lower because its assumed always in the data to be indoors and therefore simulated w/ room reverb? Likewise, one would probably expect "duck" sounds to mostly be occurring outside).

---

> ### Author Response · Authors · 2024-11-21
> **To #Ph1d (Part 1)**
>
> To #Ph1d: Thank you for your professional suggestions and comments, some of which serve as an unexpected but remarkable view of our work (especially the last paragraph).
>
> 1. Quality of the samples and simulation settings.
>     1. Simulation settings: We have strictly followed the previous research for simulation, and the difference delta is derived from the 16-18cm ear distance (Line 194) following the psychoacoustical studies (e.g., [Woodworth model](https://pmc.ncbi.nlm.nih.gov/articles/PMC3985894/)). Indeed, the stereo perception can be enhanced by simply increasing the ear distance, however, we keep using the **standard and testified distance** instead of a distinctive but unrealistic simulation setting.
>     2. Perception Difference: We show analysis of the subjective perception in Fig.3(a) and Tab.G.22, and demonstrate that statistically majority of cases follow the expected direction, despite minor fluctuations.
>     3. Tips for Listening: We kindly recommend using earphones with stereo protocol in a quiet environment to examine the demo. We would also appreciate it if you could explore the simulation cases at the bottom of the demo page and check if the problem remains.
> 2. GPT induction and manual validation.
>     * We use the GPT to efficiently generate configurations for simulation, as we expect the LLM could understand the geometry and acoustic associations by learning the "world model". The results also verify the effectiveness. Indeed, we strictly **manually validated** test sets even with multiple expert voting. As for the training set, manual checking is performed to ensure an average accuracy > 93%, as mentioned in Appendix. Sec.C and Tab.C15.
> 3. Image conditioning task and appendix (updated in Sec.3 and Sec.4).
>     * While it is true that our target and title is about language-driven generation, there is still a **general consensus** in the community that the usage of images and interactive methods to generate spatial audio is still interesting and worth exploring. Previous 1-C audio generation works including Make-An-Audio also take the language as the main modality and the other modality as the bonus. However, to enhance the clarity, we slightly reorganize the paper and emphasize the ablation in the appendix.
> 4. Critical classifier-free guidance (updated in Appendix.Sec.G.7).
>     * Thanks for the valuable advice. We add the results of **cfg scale** with the constant seed. According to the table below, cfg scale around $w=6$ is recommended to balance the trade-off.
>
>         | CFG_scale $w$   | $w=4$ | $w=5$ | $w=6$ | $w=7$ | $w=8$ | $w=9$ |
>         | --------------- | ----- | ----- | ----- | ----- | ----- | ----- |
>         | AudioCaps@CLAP↑ | 0.624 | 0.666 | 0.672 | 0.675 | 0.672 | 0.670  |
>         | AudioCaps@IS↑   | 10.63 | 13.30 | 13.79 | 13.94 | 14.13 | 13.90 |
>         | AudioCaps@FD↓   | 17.89 | 9.07  | 14.03 | 15.31 | 16.53 | 17.50 |
>         | Mix-set@FSAD↓   | 0.147 | 0.153 | 0.160 | 0.167 | 0.168 | 0.191 |
> 5. FSAD details and analysis (updated in Appendix.Sec.F.3).
>     1. Objective: FSAD processes the ITD features from a series of time slots to perceive how direction changes over time, while the GCC MAE and CRW MAE represent only the temporal average ITD. Therefore, in some moving scenarios, the effects from the left and right parts **counteract** each other.
>     2. Analysis: To demonstrate FSAD's effectiveness, we specifically simulated 1,000 moving samples for testing. In these samples, changes in the moving direction like No.1 and No.2 are not reflected by the MAEs, but perfectly reflected by the FSAD.
>         | Type | No. | Source audio     | Target audio                       | GccPHAT MAE | CRW MAE | FSAD     |
>         |------|------|-------------------------------------|-------------------------------------|-------------|---------|----------|
>         |Single Dynamic (SD)| - | 1000 samples (L to R or R to L) | 1000 reversed samples | 0.19     | 0.24 | 2.61     |
>         |Single Dynamic (SD)| 1 | A car is moving from left to right. | A car is moving from left to right. | 0.73     | 0.63 | 0.08     |
>         |Single Dynamic (SD)| 2 | A car is moving from left to right. | A car is moving from right to left. | 0.58    | 0.45 | 4.43 |
>         | Single Stationary (SS) | 3| A duck is quacking on the left. | A duck is quacking on the right. | 132.13     | 107.19 | 4.90     |

---

> ### Author Response · Authors · 2024-11-21
> **To #Ph1d (Part 2)**
>
> 6. Potential bias on categories (updated in Appendix.Sec.G.11).
>     1. Potential bias. We understand your concern about the potential bias of simulated data as a result of LLM's indication of configurations from common sense. So, we want to clarify both the capability of our SpatialSonic and the dilemma of common sense bias.
>     2. Results on different categories: In the table below, we evaluated more than 2K samples from five common classes individually and the “✓” means the successful generation. Only minor differences between them are observed, likely due to complex reasons like data scale and individual differences. Generally, **everything appears normal** except for the generation of airplane echo sounds. It seems that LLM in data collection consistently assumes that airplanes cannot exist in a small reverberating room, thereby introducing a specific type of bias.
>         |  Class | Stationary (FSAD) | Moving(FSAD) | Reverberation | Distance |
>         |--------------|-------------------|--------------|---------------|----------|
>         | Dog          | 0.141             | 0.137        | ✓   |  ✓ |
>         | Man speaking | 0.170             | 0.177        | ✓ | ✓ |
>         | Airplane        | 0.179             | 0.164        | ✗ |✓ |
>         | Piano        | 0.243             | 0.215        |✓ | ✓ |
>         | Violin       | 0.267             | 0.251        | ✓ | ✓ |
>     3. Dilemma Analysis: This potential bias from common sense reasoning is interesting and a little bit **controversial** in works like Layoutllm-T2I [1] and our work. We are trying to use the attribute “rationality” in Fig.2 to detect and exclude scenarios like “an airplane is passing by fastly in the small reverberating room”. In this case, this scenario is unrealistic and unnecessary, since an airplane is unable to fly in the small room. So, this bias could be a coin of two sides. But as a pioneer work, we still prove that most common objects do not suffer from such bias. The rest of this potential bias is more than welcome to explore in the future.
>
> *[1] Qu, L., Wu, S., Fei, H., Nie, L., & Chua, T. S. (2023, October). Layoutllm-t2i: Eliciting layout guidance from llm for text-to-image generation. In Proceedings of the 31st ACM International Conference on Multimedia (pp. 643-654).*

---

> > ### Comment · Reviewer_Ph1d · 2024-11-22
> >
> > The class-specific factuality analysis here makes sense. Thanks for investigating. I think ability to generate completely out-of-distribution phenomena is a interesting but separate topic, so agree it's fine to exclude things like "airplanes" in indoor environments in this work. I was more interested in the effect of such priors by class, e.g. "man speaking" in both indoor/outdoor+moving/not.
> >
> > (I think the Appendix.Sec.G.11 portion still needs a bit of grammar/typographic clean-up, but otherwise looks fine.)

---

> > > ### Author Response · Authors · 2024-11-23
> > > **Thanks to Reviewer #Ph1d**
> > >
> > > To #Ph1d:
> > >
> > > Thank you for your positive reply. We are glad to see that your concerns have been addressed through our factuality analysis.
> > >
> > > Apologies for the typographical error. We have reviewed Section G.11 and **corrected the grammar and spelling errors** in the latest uploaded version.
> > >
> > >  If you have any further questions, please **don't hesitate** to contact us and we will get back to you as soon as possible.
> > >
> > > Best regards

---

### Official Review · Reviewer_YTkB · 2024-11-05

**Soundness:** 3
**Presentation:** 3
**Contribution:** 3
**Rating:** 8
**Confidence:** 5

**Summary:**

The paper presents SpatialSonic, an innovative framework for generating controllable stereo audio driven by text and images. The proposed model leverages a new large-scale dataset, BEWO-1M, which includes more than 1 million stereo audio-caption pairs. SpatialSonic's one-stage approach, which incorporates azimuth-aware guidance,  addresses the challenges of multi-stage models like computational inefficiencies and error propagation. The experimental results showcase that SpatialSonic consistently outperforms existing state-of-the-art models in terms of spatial quality and subjective human evaluations.

**Strengths:**

1. Innovative Approach: The paper proposes a unique one-stage generation framework that integrates multimodal inputs (text and images) with azimuth-aware guidance, enabling controllable stereo audio synthesis.

2. Significant Dataset Contribution: The BEWO-1M dataset is a notable contribution,  providing a large-scale foundation for training and evaluating spatial audio models. Its combination of audio, captions, and simulated spatial attributes is a strong asset for advancing future research.

3. Comprehensive Methodology: The detailed explanation of the azimuth-based guidance and the fusion of multimodal embeddings make the technical approach clear and replicable.

**Weaknesses:**

1. Dependence on Simulated Data: The BEWO-1M dataset, while substantial, is built largely on simulated and GPT-transformed data. This could limit the model's ability to generalize to real-world audio data, which typically exhibits greater variability. Can you show the results on any related real-world datasets, e.g., dataset in [1]?

2. Given the synthetic nature of BEWO-1M, how do you plan to adapt or extend the dataset to include more real-world audio data? Have you considered potential data augmentation techniques or collaborative efforts to collect diverse, annotated, real-world audio?

3. Generalization Issues: The authors note the limitations of the image encoder in handling more diverse or open-world scenarios. This could impact the model's performance in applications that involve less structured inputs. What are the possible ways to eliminate it?

4. The training process requires substantial hardware resources. Do you foresee any potential strategies to reduce the computational load, such as model distillation or optimization techniques, to make the model more accessible?

5. You mentioned potential expansions involving 5.1-channel audio and higher sampling rates. What are the anticipated challenges with these enhancements, and how do you plan to overcome them?

6. The code and algorithm are not available; it is challenging to know how the model was trained.

[1]. Huang, R., Huang, J., Yang, D., Ren, Y., Liu, L., Li, M., ... & Zhao, Z. (2023, July). Make-an-audio: Text-to-audio generation with prompt-enhanced diffusion models. In International Conference on Machine Learning (pp. 13916-13932). PMLR.

**Questions:**

The final rate can be changed depending on how the weaknesses are addressed.

---

> ### Author Response · Authors · 2024-11-21
> **To #YTkB (Part 1)**
>
> To #YTkB: We appreciate very much your kind suggestions to improve our work. Following your suggestions, new experiments and ongoing extensions are illustrated here and in the Appendix for the benefit of the community.
>
> 1. Dependence on simulation data and Results on real-world data (updated in Appendix.Sec.G1).
>     * Real-world data: Although based on simulated data, our dataset and evaluation have included **adequate real data** to demonstrate generalizability. The results include the Audiocaps test set in Tab.2, the "RW-set" row in Tab.3, and additional experiments shown below with Clotho. In particular, "RW-set" is the subset of BEWO-1M annotated by our experts on real-world datasets.
>     * Supplemental experiments on Clotho: Although it is now more common to use AudioCaps (high-quality and real-world data) as the test set (including Make-An-Audio 1&2, AudioLDM 1&2, TANGO 1&2), we still manage to conduct **zero-shot experiments on Clotho**[1] (same as Make-An-Audio) to showcase the comparative performance using the popular [test codes](https://github.com/haoheliu/audioldm_eval), whose details are also updated in Sec.G.1.
>         | Model              | CLAP$\uparrow$ | FD$\downarrow$ | FAD$\downarrow$ | ISc$\uparrow$ | KL$\downarrow$ | OVL$\uparrow$ | REL$\uparrow$ |
>         | ------------------- | -------------- | -------------- | --------------- | ------------- | -------------- | ------------- | ------------- |
>         | Make-an-audio       | 0.331          | 27.32          | 6.10            | 6.94          | 3.15           | 3.27          | 3.32          |
>         | Make-an-audio2      | 0.343          | 19.10          | 3.48            | 8.19          | 2.47           | 3.56          | 3.58          |
>         | Audioldm2           | 0.340          | 25.39          | 3.49            | 7.93          | 2.62           | 3.47          | 3.48          |
>         | Tango2              | **0.363**      | 22.72          | 3.39            | 9.66          | **2.21**       | 3.53          | 3.49          |
>         | SpatialSonic (Ours) | 0.361          | **18.81**      | **3.37**        | **10.31**     | 2.36           | **3.61**      | **3.63**      |
>     * Use of Simulation: We primarily use simulated data for training, since the collection of large-scale spatial audio data with precise directional annotations in real-world settings is prohibitively expensive and time-consuming. Moreover, as demonstrated in Tab.2, Tab.3, Tab.G20, and Tab.G22, training on high-quality simulated data **actually enhances performance in real-world scenarios**. These results show the equivalent perception objective between the rigorous simulation and real-world perception.
>
> 2. Plan to adapt or extend the dataset.
>     * Yes, we have a clear plan to expand the dataset. Currently, we are using the Apple Vision Pro as the capture device, similar to [apple_support](https://support.apple.com/en-gb/guide/apple-vision-pro/dev7068c3c93/visionos). In the ideal progress, we aim to substantially extend the duration of the real-world dataset to approximately **300 hours** by Spring 2025. The updates will be released on the demo page and GitHub repository.
> 3. Image encoder Generalization
>     * We admit and expect that a better image encoder could improve the generalization. Recent spotlight works such as Open-World/Open-Vocabulary SAM [2-4] can be leveraged by our model for more general scenarios. However, since the paper has reached 40 pages, we may leave this unresolved and valuable issue for future research.
> 4. Potential strategies for computational load.
>     * We try to implement **mixed-precision training** in SpatialSonic and reduce training time by 31% and GPU memory usage by 27%. We will incorporate a [DeepSpeed](https://www.deepspeed.ai)-like mixed-precision option into the open-source training code. Besides, inference acceleration is also feasible via distillation (e.g. Latent Consistency Models [5-6]) and low-bit quantization.
> 5. Challenges to overcome 5.1-channel audio and higher sampling rates.
>     1. Challenges: The primary challenges are the **scarcity** of real-world 5.1-C data for testing and the **high training costs** associated with continuous VAEs. According to empirical conclusions, the 5.1-C VAEs will require at least 3 times the parameters and 2 times the training time.
>     2. Our effort: Our simulation can be **adapted** for 5.1-C and higher sampling rates by altering just a few lines of code. If there is a demand for 5.1-channel or higher sampling rate data, we are **prepared to release** 5.1-C or 44.1KHz open-source versions.
>     3. Contribution: Most of the popular T2A datasets (e.g., Clotho and Audiocaps) and models (e.g., Audioldm, Make-an-audio1&2, and Tango1&2) operate at 16kHz. We follow this standard for **fair comparison**. Future plans will focus on supporting the community, which does not diminish the contributions of our paper.

---

> ### Author Response · Authors · 2024-11-21
> **To #YTkB (Part 2)**
>
> 6. Unavailable code and algorithm.
>     * As outlined in Appendix Sec.D-G, we have provided extensive implementation details and valuable insights, which is uncommon in the community. Moreover, we have established the repository on GitHub and documented the Creative Commons license (CC BY) in Appendix Sec.C.4. We will make it publicly available soon **with no conditions**.
>
> *[1] Drossos, K., Lipping, S., & Virtanen, T. (2020, May). Clotho: An audio captioning dataset. In ICASSP 2020 (pp. 736-740). IEEE.*
>
> *[2] Sodano, M., Magistri, F., Nunes, L., Behley, J., & Stachniss, C. (2024). Open-World Semantic Segmentation Including Class Similarity. In Proceedings of the IEEE/CVF Conference on CVPR (pp. 3184-3194).*
>
> *[3] Liang, F., Wu, B., Dai, X., Li, K., Zhao, Y., Zhang, H., ... & Marculescu, D. (2023). Open-vocabulary semantic segmentation with mask-adapted clip. In Proceedings of the IEEE/CVF Conference on CVPR (pp. 7061-7070).*
>
> *[4] Yuan, H., Li, X., Zhou, C., Li, Y., Chen, K., & Loy, C. C. (2025). Open-vocabulary sam: Segment and recognize twenty-thousand classes interactively. In ECCV (pp. 419-437). Springer, Cham.*
>
> *[5] Song, Y., Dhariwal, P., Chen, M., & Sutskever, I. (2023). Consistency models. arXiv preprint arXiv:2303.01469.*
>
> *[6] Luo, S., Tan, Y., Huang, L., Li, J., & Zhao, H. (2023). Latent consistency models: Synthesizing high-resolution images with few-step inference. arXiv preprint arXiv:2310.04378.*

---

> ### Author Response · Authors · 2024-11-26
> **Please let us know whether all issues are addressed**
>
> Dear reviewer,
>
> Thanks for the comments. We have provided more explanations and answers to your questions. Since the deadline for discussion is near the end, please let us know whether we have answered all the questions.
>
> Best regards,
> Authors

---

> > ### Comment · Reviewer_YTkB · 2024-12-02
> > **My comments are resolved.**
> >
> > Thanks for the feedback from the authors! I would like to increase my original rate.

---

### Author Response · Authors · 2024-11-21
**Thanks to the insightful and professional reviewers.**

Dear all reviewers, we sincerely appreciate all **positive initial reviews** (8, 6, 6, 6) on our work. Your recognition of our innovative generative task, extensive dataset, and well-designed model greatly encourages us. In response to your insightful comments and valuable suggestions, we have dedicated considerable effort over the past few days to incorporate your advice into our work. Specifically, we have enhanced the following aspects in the revised draft:

* **Additional Experiment**: We have conducted further experiments to demonstrate the versatility and generalizability of our approach, including the experiment on the less commonly used **Clotho** dataset (#YTkB-Q1) and pretraining of BEWO-1M for spatial audio **retrieval** and **captioning** tasks (#pDcX-Q2).

* **Extensive Analysis**: In addition to the extensive observation and analysis provided in the previous version, we have specifically added experiments focusing on **classifier free guidance (cfg) scale** (#Ph1d-Q4), **FSAD** (#Ph1d-Q5) and potential **class-level bias** (#Ph1d-Q6).

* **Writing**: We have thoroughly revised unclear explanations (#VEpT-Q2), incorporated up-to-date references, and enhanced the paper's structure (#pDcX-Q1) to ensure better clarity and coherent organization.

Additionally, we have included a detailed revision log in **Appendix Sec.J** following this discussion. We welcome any constructive feedback to further enhance the quality of this paper, and will diligently document all subsequent revisions in the appendix.

---

### Meta-Review · Area_Chair_ZmQR · 2024-12-19

**Metareview:**

This paper proposes SpatialSonic, a spatial audio generation based on latent diffusion. The proposed model is trained by the newly proposed BEWO-1M dataset, a GPT-assisted dataset. The proposed SpatialSonic can generate stereo audio by using guidance by text instruction, image, and interactive audio generation during inference. Experimental results show the effectiveness of the proposed SpatialSonic method in terms of spatial quality and subjective human evaluations.

All the reviewers reached a positive consensus. Most of the reviewers agree that this paper has advantages in (1) the importance of stereo audio generation, (2) dataset contribution, (3) method contribution. I also agree with the reviewers' opinion. There were several weaknesses raised by the reviewers, but I think the strength of this paper outweighs its weaknesses.

**Additional Comments On Reviewer Discussion:**

There were several concerns raised by the reviewers. For example, Reviewer YTkB raised concerns regarding the dependency on synthetic data. The authors provided additional experimental results and the reviewer raised their initial score. There was a discussion related to the potential class-level bias. The authors reported additional experiments for this and Reviewer Ph1d agreed with the authors' analysis.

---

### Decision · Program_Chairs · 2025-01-22

Accept (Spotlight)